# Global meta-analysis shows action is needed to halt genetic diversity loss

Robyn E. Shaw[1,2,3,4,5,58], Katherine A. Farquharson[6,7,58], Michael W. Bruford[1,8,9,59], David J. Coates[1,2,10], Carole P. Elliott[1,2], Joachim Mergeay[1,9,11,12], Kym M. Ottewell[1,2,10], Gernot Segelbacher[1,9,13], Sean Hoban[1,9,14], Christina Hvilsom[1,9,15], Sílvia Pérez-Espona[1,16], Dainis Ruņģis[9,17], Filippos Aravanopoulos[18], Laura D. Bertola[1,19], Helena Cotrim[9,20], Karen Cox[11], Vlatka Cubric-Curik[9,21], Robert Ekblom[1,9,22], José A. Godoy[1,9,23], Maciej K. Konopiński[9,24], Linda Laikre[1,9,25], Isa-Rita M. Russo[1,8,9], Nevena Veličković[9,26], Philippine Vergeer[1,9,27], Carles Vilà[9,23], Vladimir Brajkovic[21], David L. Field[28,29], William P. Goodall-Copestake[30], Frank Hailer[1,8,31], Tara Hopley[32], Frank E. Zachos[9,33,34,35,36], Paulo C. Alves[9,37,38,39], Aleksandra Biedrzycka[9,24], Rachel M. Binks[2], Joukje Buiteveld[9,40], Elena Buzan[9,41,42], Margaret Byrne[2], Barton Huntley[2], Laura Iacolina[9,41,43,44], Naomi L. P. Keehnen[25,45], Peter Klinga[9,46,47], Alexander Kopatz[1,9,48], Sara Kurland[25,49], Jennifer A. Leonard[23], Chiara Manfrin[50], Alexis Marchesini[9,51,52], Melissa A. Millar[2], Pablo Orozco-terWengel[1,8], Jente Ottenburghs[53,54], Diana Posledovich[25], Peter B. Spencer[3,59], Nikolaos Tourvas[18], Tina Unuk Nahberger[55], Pim van Hooft[53], Rita Verbylaite[9,56], Cristiano Vernesi[1,9,57] & Catherine E. Grueber[1,6 ✉]

Mitigating loss of genetic diversity is a major global biodiversity challenge[1–4]. To meet recent international commitments to maintain genetic diversity within species[5,6], we need to understand relationships between threats, conservation management and genetic diversity change. Here we conduct a global analysis of genetic diversity change via meta-analysis of all available temporal measures of genetic diversity from more than three decades of research. We show that within-population genetic diversity is being lost over timescales likely to have been impacted by human activities, and that some conservation actions may mitigate this loss. Our dataset includes 628 species (animals, plants, fungi and chromists) across all terrestrial and most marine realms on Earth. Threats impacted two-thirds of the populations that we analysed, and less than half of the populations analysed received conservation management. Genetic diversity loss occurs globally and is a realistic prediction for many species, especially birds and mammals, in the face of threats such as land use change, disease, abiotic natural phenomena and harvesting or harassment. Conservation strategies designed to improve environmental conditions, increase population growth rates and introduce new individuals (for example, restoring connectivity or performing translocations) may maintain or even increase genetic diversity. Our findings underscore the urgent need for active, genetically informed conservation interventions to halt genetic diversity loss.

Biodiversity continues to be lost worldwide at unprecedented rates[7]. International agreements recognize biodiversity at three fundamental levels: ecosystem diversity, species diversity and within-species (intraspecific) genetic diversity (https://www.cbd.int). Intraspecific genetic diversity is critical to individual and population fitness, and thus the long-term survival of populations and species, which ensures ecosystem resilience[8,9]. Maintaining genetic diversity protects biodiversity against future environmental changes[1,10] and supports nature's contributions to society[11]. In recognition of its importance, the Convention on Biological Diversity's Kunming–Montreal Global Biodiversity Framework[12] now includes targets for safeguarding of genetic diversity of all species[5,6].

Quantification and prediction of genetic diversity change over time are essential to biodiversity policy prioritization, risk assessment and landscape management[4,12]. Population decline and fragmentation due to anthropogenic factors, such as habitat degradation, unsustainable harvest, invasive species and extreme climatic events[13–16], lead to genetic erosion[17] (loss of genome-wide genetic diversity and adaptive potential). Observed genetic diversity loss is therefore both a signal of population decline, and a conservation concern in its own right[4]. Such losses have now been reported across several taxonomic groups[18,19], and are not exclusive to rare and threatened species[13]. For example, a recent study showed around 6% loss of genetic diversity across populations of 91 animal species over the past century[13]. Theoretical predictions

based on the relationship between habitat area and genetic diversity suggest that at least 10% of genetic diversity may have already disappeared in many plant and animal species[20]. Furthermore, even greater losses are predicted on the basis of population genetic theory and the Living Planet Index, unless interventions are taken to halt and reverse species' population declines[21].

Although previous research indicates a loss of genetic diversity in specific taxonomic groups and regions[3,22], there is limited data on the extent and patterns of genetic diversity decline. Furthermore, although there is substantial evidence that individual conservation actions can have important benefits for biodiversity[23,24], there has been no temporally, spatially and taxonomically comprehensive census of genetic diversity change, alongside information about threats and management action. Although existing molecular genetic datasets can be co-analysed for this purpose (applying macrogenetics[22,25]), this can be challenging[26], prompting recent calls for greater standardization in genetic diversity reporting[27,28]. Alternatively, a comprehensive and robust assessment of the primary literature, targeting patterns and processes rather than absolute measures of population genetic diversity per se, can be conducted through statistical meta-analysis[29]. By formally combining published genetic diversity measures alongside metadata on threats and conservation actions, we can synthesize knowledge on the variables associated with population genetic diversity change.

Here we present a global meta-analysis of three decades of published data on genetic diversity change across the eukaryotic tree of life. Using meta-regressions, we quantify associations between ecological disturbance, conservation actions and genetic diversity change. We explore: (1) general patterns of genetic diversity change across varying study designs and population contexts; (2) whether greater losses are found when threats (ecological disturbance) are reported; and (3) whether there is evidence that conservation interventions can moderate (slow, halt or reverse) genetic diversity loss (aims and predictions are presented in Extended Data Fig. 1).

## A global census of genetic diversity change

Our systematic literature search identified 80,271 records, of which 882 (1.1%) met our inclusion criteria for measuring temporal genetic diversity change (that is, empirical studies of multicellular organisms that report temporal data on genetic diversity over timescales likely to have been impacted by human activities), providing 4,023 measurements for analysis (Extended Data Fig. 2, Supplementary Information 1.1 and Supplementary Data 1–3). Genetic diversity change was measured across a range of geographic regions, time frames and genetic marker types, and encompassed the eukaryotic tree of life. Publication dates spanned 34 years and 217 journals across the expected general fields of ecology, evolution, conservation and genetics, as well as narrow-focus, subject specific fields (Extended Data Fig. 3a,b, Supplementary Information 1.2 and Supplementary Data 1 and 3).

Systematic review across 141 countries representing all terrestrial and most marine realms, including 628 species from 37 classes across 16 phyla, provided a field-wide view of how genetic diversity change is measured (Fig. 1a,b, Extended Data Figs. 3b and 4a–f and Supplementary Information 1.2 and 1.3). The vast majority of species studied were animals (84.7%; comprising 59.2% vertebrates and 25.5% invertebrates), followed by plants (12.7%), fungi (1.9%) and chromists (0.6%). Most species were categorized by the International Union for the Conservation of Nature (IUCN) Red List of Threatened Species[30] as non-threatened (Least Concern, 39.3%; Near Threatened, 6.1%) or having unknown threat status (Data Deficient, 1.8%; Not Evaluated, 33.8%). One-fifth of the species were threatened (Vulnerable, 7.3%; Endangered, 6.7%; Critically Endangered, 4.9%; Extinct, 0.2%) (Fig. 1b, Extended Data Fig. 4a,b and Supplementary Information 1.3). Temporal genetic diversity change was mainly measured across nuclear or mitochondrial genomes (89.5% and 15.9% of studies, respectively) with microsatellite markers being the most common tool (Extended Data Fig. 4d and Supplementary Information 1.3), and estimated over periods of less than 1 year to 12,500 years (mean 111 years, median 6 years), for a median study midpoint of the year 2000 CE (Extended Data Fig. 4e,f and Supplementary Information 1.3).

## Genetic diversity is being lost globally

We investigated patterns of mean genetic diversity change across our dataset via Bayesian hierarchical meta-analysis, in which negative parameter estimates in our study are interpreted as a loss of genetic diversity over time, positive estimates are interpreted as a gain, and estimates close to zero suggest that genetic diversity was constant (maintained) over time. Genetic diversity change was interpreted as statistically significant when 95% highest posterior density (HPD) credible intervals did not overlap zero. For each meta-regression, parameter estimates were also compared to the model intercept (chosen as a biologically or methodologically meaningful reference category).

After sensitivity testing (Methods and Supplementary Information 1.4 and 1.5), our reduced meta-analysis dataset comprised 871 published records, providing 3,983 Hedges' $g^*$ effect sizes for modelling, encompassing 622 species from 36 classes across 16 phyla. Meta-analysis over this entire dataset revealed a small, but statistically significant loss of genetic diversity over time (Hedges' $g^*$ posterior mean = −0.11; 95% HPD credible interval −0.15, −0.07) (Fig. 2a and Supplementary Information 1.4 and 1.5). No publication bias was detected (Supplementary Information 1.5). In a few cases, extreme genetic diversity change was observed, which had detectable influence on the results; therefore, such cases were removed so that our model outputs represented the general trends present across 99% of our dataset (extreme genetic diversity changes are narrated at Supplementary Information 1.4).

Using meta-regressions, we consistently found a mean loss of genetic diversity regardless of study duration, statistical method, genetic marker type or genetic diversity metric used (Fig. 2b and Supplementary Information 1.6). The magnitude of loss varied, with greater losses detected: (1) when temporal comparisons were conducted over a long time frame (30 or more years; despite controlling for the focal species' generation length); (2) when measures were derived from linear statistical measurements (such as regression) versus comparisons of two time points or coalescent analyses; (3) when using AFLP (amplified fragment length polymorphism), haplotype, nucleotide and other data types versus microsatellite markers; and (4) when using population-level genetic diversity metrics that incorporate variant frequencies (for example, expected heterozygosity or nucleotide diversity) versus other genetic diversity metrics (Fig. 2b and Supplementary Information 1.6). Where studies reported multiple genetic diversity metrics, effect sizes were weakly or moderately correlated ($r = 0.25$–$0.55$) (Supplementary Information 1.6), suggesting that the four diversity metric types we used (variant counts, variant frequencies, individual-level diversity and effective population size; Methods) capture somewhat independent information about genetic diversity change.

From a biogeographical perspective, meta-regression showed that genetic diversity loss was observed across most terrestrial realms, which comprised a vast majority of the data (90.2%), whereas results across marine realms were more variable, albeit estimated from a small number of studies (Fig. 2c and Supplementary Information 1.7). Relative to the Palaearctic, the Arctic, Temperate Northern Atlantic and Tropical Atlantic marine realms showed significantly less genetic diversity loss, with positive parameter estimates (Fig. 2c and Supplementary Information 1.7).

From a broad evolutionary perspective, common ancestry (phylogeny) explained only a small percentage of variance in effect sizes across the dataset (3.79%) (Supplementary Information 1.5). Although patterns of genetic diversity change were not well correlated with

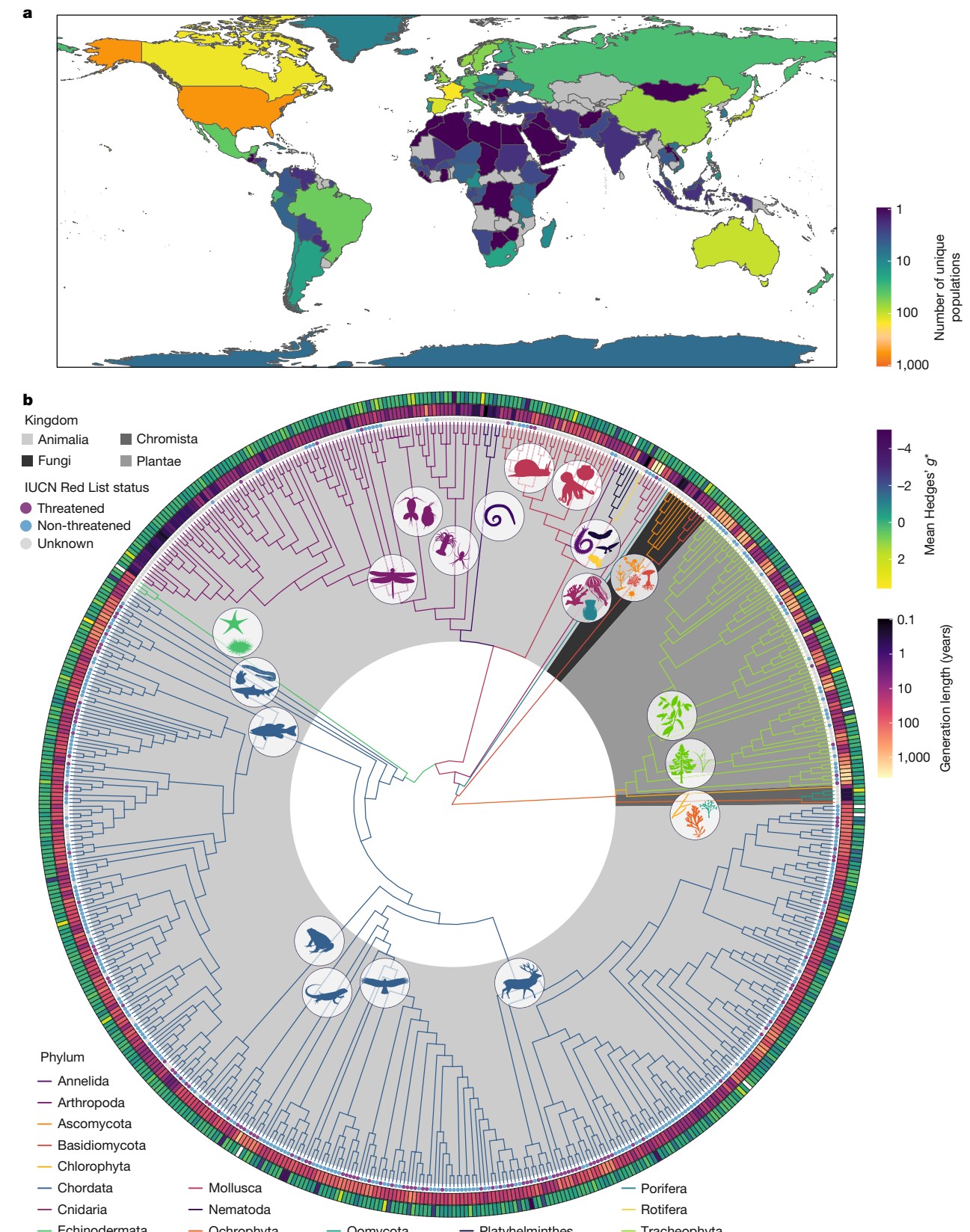

**a**

**b**

Kingdom
- Animalia
- Fungi
- Chromista
- Plantae

IUCN Red List status
- Threatened
- Non-threatened
- Unknown

Phylum
- Annelida
- Arthropoda
- Ascomycota
- Basidiomycota
- Chlorophyta
- Chordata
- Cnidaria
- Echinodermata
- Mollusca
- Nematoda
- Ochrophyta
- Oomycota
- Platyhelminthes
- Porifera
- Rotifera
- Tracheophyta

**Fig. 1 | Summary of the systematic review dataset. a**, World map with colour representing the number of unique populations included (unique species are presented in Extended Data Fig. 4). Grey represents zero counts. Note that both terrestrial and marine realms are represented within the relevant country boundaries, excluding one marine population that could not be reliably linked to a country. Studies spanning country borders are represented multiple times in this figure. World map modified from ref. 36. **b**, Visual representation of phylogenetic relationships among taxa, with IUCN Red List threat status, mean effect size (outermost ring; Hedges' $g^*$; missing data (white) represent extreme values; see Supplementary Information 1.1 and 1.4) and generation length (second outermost ring). In the tree, branch colours represent phyla, and unique classes are represented by silhouettes (coloured by phylum). Silhouettes obtained from PhyloPic (https://www.phylopic.org); image credits in Supplementary Table 15.

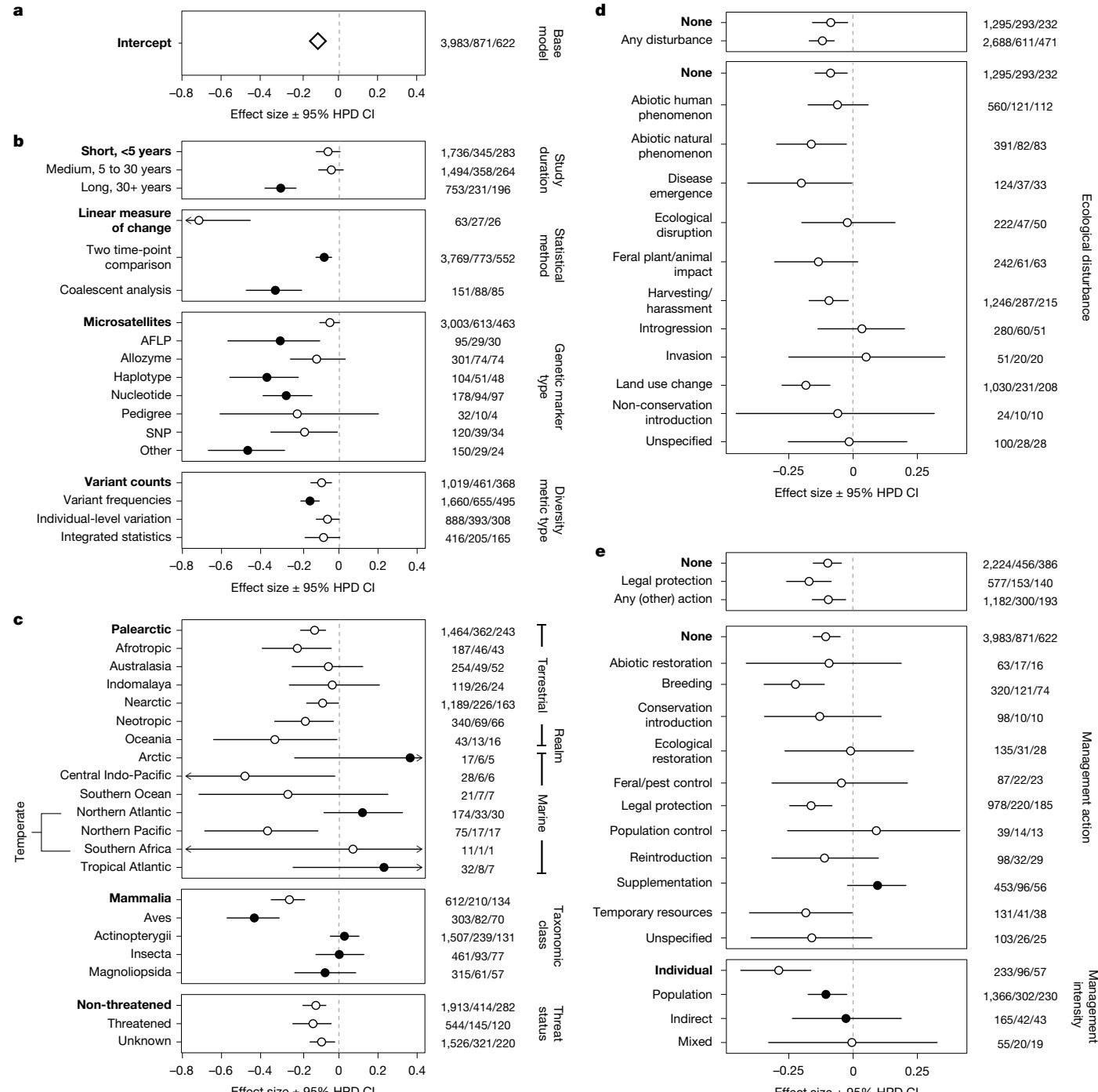

**Fig. 2 | Genetic diversity change across temporal, methodological, geographical, taxonomic, threat and conservation contexts. a–e**, Meta-regression (using the reduced meta-analysis dataset) of predicted genetic diversity change across our entire dataset (base model; **a**) and data subsets investigating associations between genetic diversity change and variables describing study design (**b**), population context (**c**), threats (**d**) and conservation management (**e**). Effect sizes (diamond for 'overall' in **a**, circles elsewhere) were measured as Hedges' $g^*$ posterior mean and error bars represent the 95% HPD credible interval (CI). A negative effect size estimate represents a loss of genetic diversity that is statistically significant if the HPD credible intervals do not overlap zero (dashed line); a positive effect size estimate represents a gain in genetic diversity that is statistically significant if the HPD credible intervals do not overlap zero. Arrows denote 95% HPD credible intervals that extend beyond axis limits. Filled circles represent predictors that are significantly different from the intercept at $\alpha = 0.05$, with the intercept for each meta-regression indicated in bold text. Numbers on the right represent sample sizes, presented as number of effect sizes/papers/species. Estimates for generation and study midpoint (included as fixed effects in all models) are provided in Supplementary Information 1.5–1.9.

ancestry relationships, variation was seen at the class taxonomic rank: of the five classes with the most data, the greatest loss of genetic diversity was observed in Aves (birds; predicted Hedges' $g^*$ posterior mean = −0.43; 95% HPD credible interval −0.57, −0.30), followed by Mammalia (mammals; Hedges' $g^*$ posterior mean = −0.25; 95% HPD credible interval −0.35, −0.17) (Fig. 2c). However, relative to Mammalia,

three taxonomic classes—Magnoliopsida (dicotyledonous plants), Insecta (insects) and Actinopterygii (ray-finned fishes)—showed significantly less loss and no significant mean genetic diversity change, suggesting that, on average, genetic diversity was maintained over time in these three taxonomic classes (Fig. 2c and Supplementary Information 1.7).

Demographic history prior to a temporal genetic study may plausibly affect our ability to detect further genetic diversity change. For those populations that were likely to have faced species-level threats and/or declines, as identified by IUCN Red List threat status[30], meta-regression showed that genetic diversity loss occurred regardless of whether a focal species was threatened, non-threatened, or had unknown threat status (Fig. 2c and Supplementary Information 1.7). We further re-examined our main findings with and without a subset of studies focused on populations identified as 'domestic, pest or pathogen' (Extended Data Fig. 5a and Supplementary Information 1.7). Greater genetic diversity losses were detected in Aves and Magnoliopsida populations in our domestic, pest or pathogen data subset, with Aves representing significantly greater genetic diversity loss relative to the model reference category, Mammalia (Extended Data Fig. 5a).

## Disturbance is more common than management

We developed and applied a protocol to categorize threats to populations (ecological disturbance, including intentional or unintentional anthropogenic events and extreme natural events; described in Extended Data Table 1), as well as conservation management actions (described in Extended Data Table 2), to quantify their effects on genetic diversity change. For those variables with sufficient data for meta-regression (Supplementary Information 1.8 and 1.9), ten types of ecological disturbance showed negligible correlations ($r \le |0.24|$), as did ten types of conservation action ($r \le |0.25|$), with the exception of a weak negative correlation between legal protection and breeding management ($r = -0.41$) (Extended Data Fig. 6 and Supplementary Information 1.10 and 1.11), suggesting that overall our categorizations provide largely independent information about threats and management.

Within the temporal time frame of studies, at least one type of ecological disturbance or conservation management action was reported for 65.11% or 45.75% of the unique populations, respectively, in our systematic review dataset, with 35.35% reporting both. For ecological disturbances, harvesting or harassment (harvesting/harassment) of the focal species was the most commonly reported disturbance (29.34%), followed by land use change (26.01%) and abiotic human phenomenon (13.56%) (Extended Data Fig. 6 and Supplementary Information 1.10). For conservation management action, legal protection (23.02%) was the most commonly reported action, followed by supplementation (adding individuals to an existing population) (10.28%) and breeding management (9.70%) (Extended Data Fig. 6 and Supplementary Information 1.11). Ecological disturbances and conservation actions occurred more commonly for threatened species (82.33% and 66.78%, respectively) compared with the non-threatened, and Data Deficient and Not Evaluated species (62.38% and 42.38%, respectively).

When comparing threatened versus non-threatened species, there were no clear trends in the types of ecological disturbance or conservation management action reported (Extended Data Fig. 6 and Supplementary Information 1.10 and 1.11). However, among taxonomic classes, ecological disturbance and conservation actions varied (Fig. 3a,b, Extended Data Fig. 7 and Supplementary Information 1.10 and 1.11). For example, harvesting/harassment, followed by land use change, were the most reported disturbances for Mammalia and Actinopterygii, with land use change ranked as the most common for Aves, Insecta and Magnoliopsida (Fig. 3a). For conservation action, other than legal

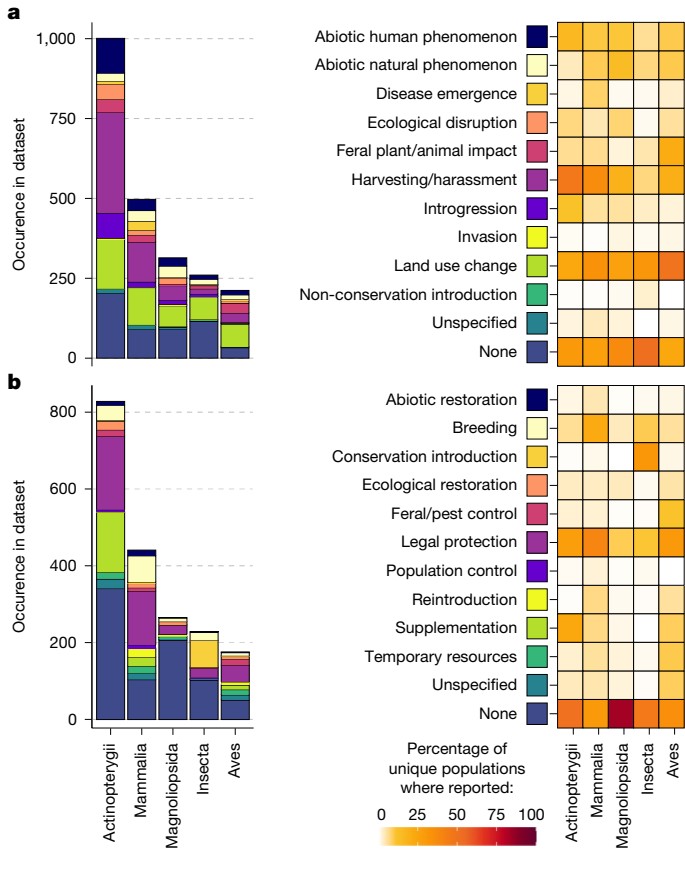

**Fig. 3 | Summary of threats and management. a,b,** Stacked bar charts showing total counts (sample size) and heat maps showing the percentage of unique populations of species for which the different types of ecological disturbance (**a**) and conservation management actions (**b**) were reported (definitions in Extended Data Tables 1 and 2 and Supplementary Information 2.5), for the five most data-rich taxonomic classes (the remaining classes are presented in Extended Data Fig. 6). Coloured squares to the right of the disturbance type and action labels indicate the colour used to represent the disturbances and actions in the bar charts.

protection (the most common action reported), the next most common action for Actinopterygii was supplementation. For Mammalia, the second most common action was breeding. For Insecta, breeding and ecological restoration were equal second most common actions. For Magnoliopsida, the second most common action was conservation introduction, and for Aves it was control of feral and pest species (Fig. 3b and Supplementary Information 1.11).

## Actions to maintain genetic diversity

Using meta-regression, we explored genetic diversity change in studies that reported ecological disturbance compared with studies in which no disturbance was reported during the time period examined. A statistically significant mean loss of genetic diversity occurred even when no disturbance was reported, suggesting a background level of genetic diversity loss across species (Fig. 2d and Supplementary Information 1.8). When disturbance types were considered individually, statistically significant genetic diversity loss was detected alongside abiotic natural disturbances (for example, wildfire), disease emergence, harvesting/harassment and land use change, although these estimates were not statistically different from background loss (that is, they did not differ from the model intercept, which

represents no reported disturbance) (Fig. 2d and Supplementary Information 1.8).

We also explored genetic diversity change in studies that reported conservation management actions during the course of the study compared with the absence of such action. Consistent with the background loss identified in our analysis of ecological disturbance, statistically significant mean loss of genetic diversity occurred in the absence of conservation actions (Fig. 2e and Supplementary Information 1.9). As legal protection alone does not involve active ecological intervention (although it may mandate it, in which case we recorded those actions if reported), we considered the effects of this action compared to all others, and found statistically significant mean loss of genetic diversity, similar to background loss (Fig. 2e, Extended Data Fig. 5b and Supplementary Information 1.9). When conservation management actions were considered individually, statistically significant genetic diversity loss was detected alongside reports of breeding, legal protection and/or temporary resources (for example, supplementary feeding), across all species and regardless of threat status (Fig. 2e, Extended Data Fig. 5b and Supplementary Information 1.9). This is not surprising, given that conservation management actions primarily target at-risk populations that may already be in decline, and such decline can result in loss of genetic diversity[31]. Even if conservation management actions succeed in slowing, halting or reversing genetic diversity decline, a net genetic diversity loss may still be recorded (Extended Data Fig. 1). By contrast, mean estimates for genetic diversity change were close to zero or positive when reported alongside ecological restoration, feral and pest control, population control and supplementation, suggesting that, on average, genetic diversity was maintained or increased across temporal comparisons. Supplementation was a statistically significant moderator of genetic diversity loss, and was the only conservation action associated with a significant increase in genetic diversity compared with cases in which no action was reported, especially in birds (Fig. 2e and Extended Data Fig. 5c). The positive effect of supplementation was observed in non-threatened species, but not in species that were threatened or had unknown threat status (Extended Data Fig. 5b and Supplementary Information 1.9). Considering conservation actions for the five most data-rich classes, loss of genetic diversity was observed in the absence of conservation action for Mammalia and Aves, but not for Actinopterygii, Insecta or Magnoliopsida (Extended Data Fig. 5c and Supplementary Information 1.9).

We classified conservation interventions into three levels of management intensity—namely actions that target individuals, populations or landscapes. The greatest loss of genetic diversity was associated with reports of the highest management intensity (that is, management at the individual level) (Fig. 2e and Supplementary Information 1.9). Compared with individual-level conservation management, significantly less loss was observed alongside studies reporting population-level management (for example, habitat restoration), but there was still an overall net loss of genetic diversity. Contexts associated with indirect management (for example, management targeting other species in the same habitat) showed no mean genetic diversity change, but this estimate had low precision (Fig. 2e and Supplementary Information 1.9).

## Discussion

Here we report an overall global decline in intraspecific genetic diversity. Our study provides the most comprehensive investigation of within-population genetic diversity change to date, transcending taxonomic and geographic boundaries, and the a priori objectives, predictions and methods of individual biological research reports. In birds and mammals in particular, the evidence for genetic diversity decline is clear. In other taxa, for which we had sufficient data (dicotyledonous plants, insects and ray-finned fishes), genetic diversity was maintained over time. However, these taxonomic groups may still be at risk, as genetic diversity losses are not always easily detected (Extended Data Fig. 1) or may lag behind demographic changes[32]. Declines in census sizes of species with massive populations or very long-lived species might not lead to measurable losses of genetic diversity over the timescales studied. Our finding of significant losses of genetic diversity across short study periods (on average) for several taxonomic classes, representing 207 species (with even more trending negative, although non-significant), indicates that the population size declines underlying these genetic diversity losses are likely to be considerable. This pattern carries two key implications: (1) further genetic diversity loss in the near term is likely if human societies do not take action urgently; and (2) we currently have sensitive methods and datasets for detection of genetic diversity change, which enable us to target biodiversity conservation actions effectively.

Most of the unique populations in our dataset were reportedly affected by disturbances within the time frame of the study, suggesting that anthropogenic activities are direct and widespread hazards, affecting not only diversity among species[3,33], but also genetic diversity within species. For these reasons, even the subtle negative trends of genetic diversity change that we report here should raise concern over the resilience of populations and the capacity for natural ecosystems to sustain vital ecosystem services[11], and should therefore trigger intensified conservation management actions to halt genetic erosion before further losses occur. Genetic diversity accumulates over evolutionary timescales through mutation and once lost, is difficult to restore[34]. Supplementation (that is, the addition of individuals to a population, including genetic or demographic rescue through restoring connectivity or performing translocations) was the only conservation management action associated with a statistically significant mean increase in genetic diversity over time relative to cases where no actions were reported. In addition, we found that other conservation actions designed to improve environmental conditions and increase population growth rates may halt or reduce further genetic diversity loss and therefore safeguard it. We have four recommendations to track within-population genetic diversity and determine when and where conservation actions may be needed:

(1) *Conduct temporal genetic monitoring*. Genetic diversity metrics are sensitive to change, particularly over long-term studies. Monitoring genetic diversity alongside threats and conservation action can inform strategic management.

(2) *Where temporal genetic data do not exist, start collecting now*. Although multiple-timepoint sampling informs change, single snapshots of genetic diversity are invaluable for tailoring management decisions and provide a point of comparison for future sampling.

(3) *Where genetic data collection is difficult, utilize existing data*. Here we have identified hundreds of datasets as a starting point for informing genetic management to expand upon in the future.

(4) *Where genetic data are absent, use proxies*. Genetic considerations should inform any biodiversity risk assessment, even if based solely on other data types, such as field observations of population size[35].

Our analysis demonstrates that genetic diversity loss is a realistic general expectation for many species around the world. However, we also show that we have the theoretical and technical means, as well as the on-ground conservation management approaches, to prevent further loss if we act now.

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

[1]International Union for the Conservation of Nature (IUCN) Conservation Genetics Specialist Group (CGSG), https://www.cgsg.uni-freiburg.de/. [2]Biodiversity and Conservation Science, Department of Biodiversity, Conservation and Attractions, Kensington, Western Australia, Australia. [3]School of Environmental and Conservation Sciences, Murdoch University, Perth, Western Australia, Australia. [4]Division of Ecology and Evolution, Research School of Biology, The Australian National University, Canberra, Australian Capital Territory, Australia. [5]Centre for Conservation Ecology and Genomics, University of Canberra, Canberra, Australian Capital Territory, Australia. [6]School of Life and Environmental Sciences, Faculty of Science, The University of Sydney, Camperdown, New South Wales, Australia. [7]Australian Research Council Centre of Excellence for Innovations in Peptide and Protein Science, The University of Sydney, Camperdown, New South Wales, Australia. [8]School of Biosciences, Museum Avenue, Cardiff University, Cardiff, UK. [9]European Cooperation in Science and Technology (COST), COST Action CA 18134 'Genomic Biodiversity Knowledge for Resilient Ecosystems (G-BiKE)', https://www.cost.eu/actions/ca18134/. [10]School of Biological Sciences, The University of Western Australia, Crawley, Western Australia, Australia. [11]Research Institute for Nature and Forest, Geraardsbergen, Belgium. [12]Ecology, Evolution and Biodiversity Conservation, KU Leuven, Leuven, Belgium. [13]Wildlife Ecology and Management, University Freiburg, Freiburg, Germany. [14]The Center for Tree Science, The Morton Arboretum, Lisle, IL, USA. [15]Copenhagen Zoo, Frederiksberg, Denmark. [16]The Royal (Dick) School of Veterinary Studies and The Roslin Institute, The University of Edinburgh, Easter Bush Campus, Midlothian, UK. [17]Genetic Resource Centre, Latvian State Forest Research Institute "Silava", Salaspils, Latvia. [18]Laboratory of Forest Genetics and Tree Breeding, Faculty of Agriculture, Forestry and Natural Environment, Aristotle University of Thessaloniki, Thessaloniki, Greece. [19]Department of Biology, University of Copenhagen, Copenhagen, Denmark. [20]cE3c—Center for Ecology, Evolution and Environmental Change and CHANGE—Global Change and Sustainability Institute, Faculdade de Ciências, Universidade de Lisboa, Lisboa, Portugal. [21]Department of Animal Science, University of Zagreb Faculty of Agriculture, Zagreb, Croatia. [22]Wildlife Analysis Unit, Swedish Environmental Protection Agency, Stockholm, Sweden. [23]Estación Biológica de Doñana (EBD-CSIC), Seville, Spain. [24]Institute of Nature Conservation, Polish Academy of Sciences, Kraków, Poland. [25]Department of Zoology, Division of Population Genetics, Stockholm University, Stockholm, Sweden. [26]Department of Biology and Ecology, Faculty of Sciences, University of Novi Sad, Novi Sad, Serbia. [27]Plant Ecology and Nature Conservation Group, Wageningen University, Wageningen, The Netherlands. [28]Applied BioSciences, Macquarie University, Sydney, New South Wales, Australia. [29]School of Science, Edith Cowan University, Joondalup, Western Australia, Australia. [30]Royal Botanic Garden Edinburgh, Edinburgh, UK. [31]Institute of Zoology Joint Laboratory for Biocomplexity Research (CIBR), Chinese Academy of Sciences, Beijing, China. [32]Royal Botanic Gardens Victoria, Melbourne, Victoria, Australia. [33]Natural History Museum Vienna, Vienna, Austria. [34]Department of Evolutionary Biology, University of Vienna, Vienna, Austria. [35]Department of Genetics, University of the Free State, Bloemfontein, South Africa. [36]Research Institute for the Environment and Livelihoods, Charles Darwin University, Casuarina, Northern Territory, Australia. [37]CIBIO, Centro de Investigação em Biodiversidade e Recursos Genéticos, InBIO/ BIOPOLIS Program in Genomics, Biodiversity and Land Planning, University of Porto, Porto, Portugal. [38]Department of Biology, Faculty of Sciences, University of Porto, Porto, Portugal. [39]EBM, Biological Station of Mértola, Mértola, Portugal. [40]Centre for Genetic Resources, The Netherlands, Wageningen University, Wageningen, The Netherlands. [41]Faculty of Mathematics, Natural Sciences and Information Technologies, University of Primorska, Koper, Slovenia. [42]Faculty of Environmental Protection, Velenje, Slovenia. [43]Department of Veterinary Medicine, University of Sassari, Sassari, Italy. [44]Department of Chemistry and Bioscience, Aalborg University, Aalborg, Denmark. [45]Department of Ecology, Swedish University of Agricultural Sciences, Uppsala, Sweden. [46]Department of Phytology, Faculty of Forestry, Technical University in Zvolen, Zvolen, Slovakia. [47]Department of Forest Ecology, Faculty of Forestry and Wood Sciences, Czech University of Life Sciences Prague, Prague, Czech Republic. [48]Norwegian Institute for Nature Research (NINA), Trondheim, Norway. [49]Department of Earth Sciences, Natural Resources and Sustainable Development, Uppsala University, Uppsala, Sweden. [50]Department of Life Sciences, University of Trieste, Trieste, Italy. [51]Research Institute on Terrestrial Ecosystems (IRET), The National Research Council of Italy (CNR), Porano, Italy. [52]National Biodiversity Future Center, Palermo, Italy. [53]Wildlife Ecology and Conservation Group, Wageningen University, Wageningen, The Netherlands. [54]Forest Ecology and Forest Management, Wageningen University, Wageningen, The Netherlands. [55]Slovenian Forestry Institute, Ljubljana, Slovenia. [56]Department of Forest Genetics and Tree Breeding, Institute of Forestry, Lithuanian Research Centre for Agriculture and Forestry, Kédainiai, Lithuania. [57]Forest Ecology Unit, Research and Innovation Centre, Fondazione Edmund Mach, San Michele all'Adige, Italy. [58]These authors contributed equally: Robyn E. Shaw, Katherine A. Farquharson. [59]Deceased: Michael W. Bruford, Peter B. Spencer. ✉e-mail: catherine.grueber@sydney.edu.au

## Methods

### Compilation of genetic diversity change measures

We conducted a literature search of peer-reviewed publications using the Web of Science (WOS) advanced search functions to find published works that contained temporal measures of genetic diversity. Our search was intentionally broad and followed established preferred reporting items for systematic reviews and meta-analysis (PRISMA) protocols as closely as possible[37,38]. The online search was conducted on 18 January 2019 using keywords targeting population genetic measurements, regardless of the direction of change (for example, 'increase' and 'decrease' were both included as search terms) (Supplementary Information 2.1 and Supplementary Data 4). A total of 80,271 records were retrieved, 78,727 after duplicate removal. We obtained full texts for 70,069 of these records. The remaining records were screened manually via their titles, abstracts and keywords, of which 8,596 were excluded against our inclusion criteria; 62 full texts could not be obtained (Extended Data Fig. 2 and Supplementary Information 2.1). We then performed full text mining in R v.3.5.2[39], using the packages pdfsearch v.0.2.3[40], dplyr v.0.8.0[41], and stringi v.1.3.1[42] to remove records that did not contain population genetic keywords (Supplementary Information 2.2). This resulted in 34,346 putatively includable studies of genetic diversity change, which were classified into thematic clusters using the package revtools v.0.4.0[43] (Supplementary Information 2.2).

We performed initial screening and data extraction for all 34,346 works, followed by a series of re-extraction and data validation steps. Manual screening of studies against inclusion criteria, and extraction of genetic diversity measurements and metadata took place simultaneously by members of the authorship team, via multiple workshops using shared written guidelines (Extended Data Tables 1 and 2 and Supplementary Information 2.3–2.6). Studies were suitable for inclusion in our analysis only if they satisfied all of the following criteria:

- The research must report primary, quantitative, empirical data from a multicellular nonhuman organism.
- Laboratory and experimentally manipulated populations were excluded, where these experimental manipulations were for the purpose of testing a hypothesis related to population demography or genetics (note that populations established in controlled conditions for supportive breeding or propagation were potentially includable, such as 'captive' or 'agricultural' populations),
- The time frame of the study plausibly took place over timescales likely to have been impacted by human activities, regardless of whether the study organism was actually impacted by human activities (in general, we targeted genetic diversity changes in the last few hundred years and excluded studies on ancient admixture or expansion in response to events on 'geological' timescales; further detail in Supplementary Information 2.4),
- The study design enabled a temporal comparison of population genetic measurements (for example, samples collected over multiple years) or inference (for example, coalescent genetic studies),
- The study reports a quantitative measurement of within-population 'genetic diversity' (broadly defined), and an associated measurement error and sample size. Genetic diversity statistics were obtained from main texts (including tables and figures) and supplementary materials, but no re-analysis of published datasets was conducted. Summary statistics (mean and s.d.) were calculated from tabulated data where available.

In addition to recording bibliographic data for each record, we extracted data that would enable us to calculate our effect sizes (see below and Supplementary Information 2.4), as well as corresponding metadata for meta-regression (Extended Data Tables 1 and 2 and Supplementary Information 2.5 and 2.6). Our dataset included many studies for which the main goal was not an assessment of genetic diversity or change in genetic diversity per se, but which nevertheless reported temporal measures of genetic diversity that otherwise met the inclusion criteria for our meta-analysis.

We captured genetic diversity statistics aligned with three possible study designs (see Supplementary Information 2.4): (1) linear measure of change (for example, regression of two or more time points), yielding primarily statistical measurements, such as regression coefficients or $t$ statistics; (2) two timepoint comparison (for example, comparison of two means), yielding primarily pairs of mean diversity estimates; or (3) coalescent analysis, yielding either statistical or genetic measurements, obtained by probabilistic modelling of past effective population sizes using a single sample in time[44,45]. The latter were uniquely identified in our analysis due to important differences in the underpinning theoretical framework for coalescent analyses. That is, 'early' measures of genetic diversity are not taken from real biological samples, but instead inferred from recent data and principles of genetic inheritance. Further, early time points in coalescent analyses may be identified by authors a priori based on environmental or other non-genetic hypotheses, or post hoc on the basis of substantive patterns in the data.

Genetic diversity change data were recorded alongside corresponding error estimates and sample sizes; all were extracted using the same level of precision as reported in the paper. We also recorded the time frame of the study (early and recent years, used to calculate study duration and study midpoint), plus amount and type of genetic data used (for example, number of loci, genetic marker type, genome). We classified genetic diversity change data into four metric types, aligned with 'essential biodiversity variables' for genetic composition[28]: (1) variant counts (for example, allelic richness); (2) evenness of variant frequencies (for example, expected heterozygosity and nucleotide diversity); (3) population means of individual-level variation (for example, observed heterozygosity and pedigree inbreeding); and (4) integrated statistics (for example, effective population size; see below).

We were particularly interested in associations between genetic diversity change and ecological disturbance or conservation management action, so these 'impact metadata' were collected where threats and/or conservation management actions were reported in a paper as plausibly impacting the study population between the sampling time points of the study. In principle, we categorized ecological disturbances as events with potential to impair conditions for the focal species or its habitat, and conservation management actions as human activities intended to improve conditions for the focal species or its habitat. For the latter, we also considered the intensity of conservation management actions (that is, the magnitude of conservation intervention as probably experienced by the focal species). We also collected additional metadata, as the objectives of ecological disturbances are likely to vary across species. For example, disturbance is often intentional for pests and pathogens (for example, population reduction), whereas disturbance of threatened species can include indirect or unintentional consequences of human activity (for example, habitat loss and fragmentation). Brief definitions of the categories that we used for each of these variables are in Extended Data Table 1 and 2 and full definitions are provided in Supplementary Information 2.5.

Additional moderators that were collected included (Supplementary Information 2.6): taxonomic identity of study species (nomenclature standardized by literature review to align with Open Tree of Life[46]), country and terrestrial and/or marine realm of the locality where samples were collected, following refs. 36,47,48, along with unique site identifiers in the case of multiple populations reported in a publication. We also collected the threat status of the study species[30]; generation length of the study species (Supplementary Data 5); and classification of domesticated species or populations considered as pathogens or pests (based on description in the source publication, relevant databases or other published sources).

Many studies reported multiple measurements of genetic diversity that were suitable for inclusion in our analysis. We extracted independent measures of genetic diversity change per publication taking

into consideration the sampling scheme of the reported study and analysis of data subsets (Supplementary Information 2.4). Procedures for controlling non-independent data, missing data, infinite confidence intervals (in estimates of effective population size), zero variances, and unconventional study designs are described at Supplementary Information 2.7–2.9. After initial extraction, all included studies were re-processed by at least two additional members of a small validation group from the authorship team, to ensure consistency in the collection of genetic and metadata (Supplementary Information 2.10).

## Systematic review

Full bibliographic details for each included study were automatically downloaded from the WOS during the original search (see also Supplementary Data 1–3). Additional bibliographic metadata were also collected from the WOS, including journal title abbreviations, WOS subject categories for which each journal ranked highest, and the impact factor percentile ranking for each journal within its WOS category for 2020. Publication trends and the characteristics of studies included in our final dataset were summarized visually using the R packages ggplot2 v.3.4.3[49], treemapify v.2.5.5[50] and ggridges v.0.5.4[51]. We also explored patterns of co-occurrence and characterized the variation of ecological disturbance and conservation management actions reported across our full dataset, and data subsets of the five most data-rich taxonomic classes. We calculated pairwise Spearman's rank correlation coefficients ($r$) among ecological disturbance categories, and conservation management actions, and visualized results in R using corrplot v.0.92[52]. We also examined relationships among these variables using principal component analysis, visualized with the package factoextra v.1.0.7[53].

## Phylogeny

To visualize the taxonomic diversity of species in our dataset and their evolutionary relationships, we generated a phylogenetic tree using the R package rotl v.3.0.12[54] using Open Tree of Life IDs as described above. *Saccharomyces cf. cerevisiae* (ott id 7511391) was used as the outgroup. Six species could not be placed in the phylogeny due to unresolved taxonomy: the Japanese mud snail (*Batillaria attramentaria*), white seabream (*Diplodus sargus*), a fruit fly (*Drosophila pseudoobscura*), a sea snail (*Euparthenia bulinea*), fourfinger threadfin (*Eleutheronema tetradactylum*) and the bicolour damselfish (*Stegastes partitus*). The phylogeny was visualized using the R packages ggtree v.3.8.2[55], ggtreeExtra v.1.10.0[56], ggimage v.0.3.3[57] and rphylopic v.1.2.1[58]. Silhouettes of representative organisms for each taxonomic class were downloaded from PhyloPic (https://www.phylopic.org; see Supplementary Information 2.6 for credits). Owing to the taxonomic diversity of species in our study, obtaining a dated tree across all species was not possible and so the topology of the tree was used in modelling.

## Effect size extraction and calculation

For each comparison that satisfied our inclusion criteria, we calculated Hedges' $g^*$ (sometimes referred to as Hedges' $d^{59}$ with sample size correction $J$) as our measure of effect size. Hedges' $g^*$ was selected as the effect size measure as it is based on the standardized mean difference between two values, in our case the 'early' and 'recent' time points, minimizes over-inflation of effect size estimation in studies with sample sizes <20, and outperforms other common effect size measures such as Cohen's $d$ and Glass' $\Delta$ when the assumption of homogeneity of variance is violated[60]. All formulae used to evaluate Hedges' $g^*$ and its error are reported in Supplementary Information 2.11.

Calculation of Hedges' $g^*$ requires the sample size and error associated with the measure of genetic diversity change. Depending on the way in which genetic diversity metrics or their summary statistics are calculated, the associated sample size for effect size calculation may be for example, the number of loci, the number of samples, the number of populations or a rarefied sample size. For comparisons based on linear

measures of genetic diversity change, which varied in the methods used to determine genetic change, each paper was manually checked to retrieve the appropriate sample size and error. For two timepoint comparisons and comparisons based on coalescent analyses, we followed a hierarchical procedure to establish the sample size to use for each effect size (Supplementary Information 2.11). Multiple error types were reported (for example, s.d. or confidence intervals), and so Hedges' $g^*$ was calculated using published formulae for interconversion of these data types (Supplementary Information 2.11).

For comparisons where an effect size was calculated, the direction of the effect was determined. We ensured consistent directionality among the following measures of genetic diversity (note that many metrics were recorded in our dataset, and so the abbreviations reported below are summaries only):

(1) Variant counts were all positively associated with genetic diversity: mean of alleles across loci ($A$), standardized by sample size ($A_R$), sum of alleles across loci ($T_A$), total number of private alleles (pA) and number of polymorphic loci (NPL).

(2) Variant frequencies:
   a. Positive: expected heterozygosity ($H_E$), nucleotide diversity ($\pi$), haplotype diversity ($h$), Shannon diversity index ($H$), polymorphic information content (PIC), number of effective alleles (NEA), frequency of an allele of interest (Freq) and mean individual nucleotide p-distance (NPD, occasionally seen in major histocompatibility complex and similar studies).
   b. Negative: population-level inbreeding coefficient or selfing/outcrossing rate ($F_{IS}$), mean relatedness or kinship among individuals ($R$), band sharing score (BS) and among-population $F_{ST}$ (ap$F_{ST}$).

(3) Individual-level diversity measures:
   a. Positive: observed heterozygosity ($H_O$), standardized observed heterozygosity (SH) and mean number of alleles per individual ($A_i$).
   b. Negative: individual-level inbreeding coefficient or coancestry ($F$).

(4) Integrated statistics were all positively correlated with genetic diversity: effective population size ($N_e$), effective number of breeders ($N_b$), female effective population size ($N_f$), effective population size estimated from demographic data ($N_d$), effective population size estimated from a population census, and calculated based on an assumption about the ratio between effective and census population sizes ($N_c$).

For comparisons where genetic diversity metric type was recorded as 'other', each paper was manually checked to determine the correct direction of the effect given the context of the metric within the publication and the authors' interpretation of genetic diversity change as a loss or gain. For negatively correlated metrics, we multiplied the Hedges' $g^*$ effect size by −1 to reverse the direction of the effect—that is, across our dataset a positive Hedges' $g^*$ represents an increase in genetic diversity and a negative Hedges' $g^*$ represents a loss of genetic diversity.

All calculated effect sizes >|4| were manually examined as potential outliers by a single member of the research team, to confirm absence of data entry errors. Considering our wide diversity of statistics, these data were also checked for possible calculation errors, misinterpretation of statistical error (for example, standard error versus s.d.), or other discrepancies. Results of screening of extreme values can be found in Supplementary Information 1.4.

## Meta-analysis

We fitted multi-level Bayesian hierarchical models in the R package MCMCglmm v.2.34[61], with paper ID as a random effect for all models to account for non-independence introduced by studies that report multiple, includable effect sizes. Genetic diversity change was modelled per generation by including the $z$-standardized number of generations for that species (that is, number of years passed between the early

and recent time points, divided by generation length) as a fixed effect in all meta-regressions. Unless otherwise stated in Supplementary Information 2.12, all meta-regressions also included a fixed effect of the $z$-standardized study midpoint (year).

Each model was run for 6,000,000 iterations, with a burn-in of 200,000 and a thinning interval of 5,000, using the weakly informative inverse-gamma prior. We report the posterior mean and the 95% HPD credible intervals for each model set. Estimates with a 95% HPD credible interval excluding zero were considered statistically significant at $\alpha = 0.05$. Model diagnostics were visually checked for no pattern in the trace plots; effective size >1,000 and autocorrelation <0.1 between lag points were both checked using coda v.0.19.4[62]. Chain convergence was confirmed by passing the Heidelberger and Welch's half-width and stationarity tests in coda. Additionally, each model was independently run three times to calculate a Gelman-Rubin convergence diagnostic of <1.1 using the potential scale reduction factor. The deviance information criterion (DIC) was obtained for each of the three models, and the model with the lowest DIC was selected for interpretation.

Our base model included fixed and random effects described above (that is, fixed effects = $z$-standardized midpoint, $z$-standardized number of generations; random effect = paper ID), although variations of this model underwent sensitivity testing to determine the influence of including phylogeny as an additional random effect, and including extreme values in the dataset (described in Supplementary Information 2.11). The extended heterogeneity statistic[63] was calculated for both the base model and the sensitivity testing model that included phylogeny. Extended heterogeneity statistics partition total heterogeneity ($I^2_{total}$) into phylogenetic variance (in the phylogenetic model, $I^2_{phylogeny}$), study ID variance ($I^2_{study}$) and residual variance[63,64] ($I^2_{residual}$). For the phylogenetic model, we also obtained lambda (phylogenetic signal ($H^2$)) as the variance of the random effect of phylogeny divided by the total variance of all random effects (phylogeny, study ID and residual variance). Total heterogeneity was high, but phylogenetic signal only explained 5.48% of overall variance, so was excluded from further modelling (see also Supplementary Information 1.5); this also allowed for simplification of the model structure. Based on the results of the sensitivity testing (Supplementary Information 1.5), we excluded phylogeny and extreme values from subsequent meta-analytic modelling.

We assessed publication bias in our meta-analysis using two methods. First, we investigated time-lag bias, where different patterns in genetic diversity change may be reported over the years of publication. Such bias may plausibly occur given that methods for measuring genetic diversity have advanced substantially in recent decades. Therefore, we fitted the final base model with the addition of a $z$-standardized year of publication fixed effect. Evidence of time-lag bias is inferred if the 95% HPD credible interval of the slope estimate excludes zero. Second, we plotted Hedges' $g^*$ precision against Hedges' $g^*$ in a funnel plot to visualize possible publication bias that can occur if, for example, smaller studies without statistically significant results are not published. We did not observe time-lag bias nor funnel plot asymmetry (Supplementary Information 1.5). Given the high heterogeneity and lack of detectable publication bias in our dataset, we proceeded with meta-regression modelling.

Meta-regressions were conducted to assess the impact of different moderator variables on genetic diversity change. These were broadly categorized into moderators related to: (1) how genetic diversity change is measured (that is, study design); (2) where genetic diversity change is measured and in what species (that is, population context); (3) ecological disturbances (that is, threats); and (4) conservation interventions (that is, conservation management). In meta-regression, the coefficients estimate how each category differs from the nominated reference group, represented by the intercept[65]. As all moderator variables were categorical, we performed separate meta-regressions for each moderator to avoid the confounding effects of correlations and allow for

biologically meaningful interpretation of categorical variables relative to the intercept. For all models, moderator variables were only included if there were 10 or more effect sizes contributing to a category[65]. All models were run with the weakly informative inverse-gamma prior, the paper ID random effect and the standardized year midpoint and the number of generations over which the study took place (as a measure of study length) as fixed effects (unless otherwise specified), and additional fixed effects described in Supplementary Information 2.12.

## Inclusion and ethics statement

No ethical approval or guidance was required as data were collected only from previous studies.

## Reporting summary

Further information on research design is available in the Nature Portfolio Reporting Summary linked to this article.

## Data availability

All datasets associated with this paper are available on Zenodo: https://doi.org/10.5281/zenodo.13903787 (ref. 66). The full bibliography of 882 included papers (including their DOIs) is provided in Supplementary Data 1. We used publicly available databases to obtain species characteristics for the 628 species included in our study. Full methods are in Supplementary Information 2.6. Generation lengths (Supplementary Data 5) were obtained from scientific literature and databases including Search FishBase (https://www.fishbase.se/search.php), AmphibiaWeb (https://www.amphibiaweb.org) and CABI Compendium (https://www.cabidigitallibrary.org/journal/cabicompendium). Threat status was sourced from the IUCN Red List of Threatened Species[30] during June to August 2021. Invasive species status was sourced from the IUCN 100 of the World's Worst Invasive Alien Species list (https://www.iucngisd.org/gisd/100_worst.php). Pathogen and pest statuses were sourced from the scientific literature and databases including the European and Mediterranean Plant Protection Organization Global Database (https://gd.eppo.int/), The Global Pest and Disease Database (www.gpdd.info), and CABI Compendium (https://www.cabi.org/isc).

## Code availability

Custom text mining code is available on Zenodo: https://doi.org/10.5281/zenodo.13903787 (ref. 66).

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

**Acknowledgements** R.E.S. acknowledges funding from an Australian Research Council Linkage Project (LP170100061). K.A.F. acknowledges funding from the Australian Research Council Centre of Excellence for Innovations in Peptide and Protein Science (CE20010012). V.C.-C. acknowledges funding support from the Croatian Science Foundation (grant IP-2018-01-8708). R.E. acknowledges funding support from Uppsala University. A.B. and M.K.K. acknowledge funding support from Institute of Nature Conservation, Polish Academy of Sciences. L.L. acknowledges funding support from the Swedish Research Council Formas (grant 2020-01290) and the Swedish Research Council (grant 2019-05503). F.H. was supported by a Royal Society International Exchanges grant (IEC\NSFC\181744). J.B. acknowledges funding support from the Fundamental Research Programme 'Circular and Climate Neutral' (KB-34-013-003) funded by the Dutch Ministry of Agriculture, Fisheries, Food Security and Nature. E.B. acknowledges funding support from Slovenian Research and Innovation Agency (programme group P1-0386 and project N1-0281). L.I. acknowledges funding support from Slovenian Research and Innovation Agency (programme group P1-0314). N.L.P.K. was supported by funding to L.L. from the Swedish Research Council Formas (grant 2020-01290) and the Swedish Research Council (grant 2019-05503). P.K. was supported from the Scientific Grant Agency VEGA (1/0328/22). A.K. was supported by Norges forskningsråd (the Research Council of Norway) (project no. 160022/F40 NINA). S.K. was supported by funding to L.L. from the Swedish Research Council Formas (grant 2020-01290) and the Swedish Research Council (grant 2019-05503). A.M. acknowledges the support of NBFC, funded by the Italian Ministry of University and Research, P.N.R.R., Missione 4 Componente 2, "Dalla ricerca all'impresa", Investimento 1.4, Project CN00000033. D.P. was supported by funding to L.L. from the Swedish Research Council Formas (grant 2020-01290) and the Swedish Research Council (grant 2019-05503). F.A., W.P.G.-C., E.B. and J.A.L. acknowledge the support of the Horizon Europe Framework Programme of the European Union under grant agreement 101059492 (Biodiversity Genomics Europe). T.U.N. was financed through ARIS Research Group P4-0107. C.E.G. acknowledges funding support from a University of Sydney Robinson Fellowship. We thank the undergraduate students of M.W.B. at Cardiff University who participated in earlier parts of this project. This Article is based on work from COST Action G-BiKE, CA 18134, supported by COST (European Cooperation in Science and Technology; https://www.cost.eu). We thank Tovetorp Research Station, Stockholm University, for hosting the initial project workshop. Further information on research design is available in the Nature Portfolio Reporting summary linked to this article.

**Author contributions** Study conception: R.E.S., K.A.F., M.W.B., D.J.C., C.P.E., J.M., K.M.O., G.S., S.H. and C.E.G. Study design (methods testing): R.E.S., K.A.F., D.J.C., C.P.E., J.M., K.M.O., G.S., C.H., S.P.-E., D.R., F.A., L.D.B., V.C.-C., R.E., J.A.G., M.K.K., L.L., P.V., C. Vilà and C.E.G. Study design (funding and workshop): G.S., L.L. and C.E.G. Data acquisition (collection): R.E.S., K.A.F., M.W.B., D.J.C., C.P.E., J.M., K.M.O., G.S., S.H., C.H., S.P.-E., D.R., F.A., L.D.B., H.C., K.C., V.C.-C., R.E., J.A.G., M.K.K., L.L., I.-R.M.R., N.V., P.V., C. Vilà, V.B., D.L.F., W.P.G.-C., F.H., T.H., F.E.Z., P.C.A., A.B., R.M.B., J.B., E.B., M.B., L.I., N.L.P.K., P.K., A.K., S.K., J.A.L., C.M., A.M., M.A.M., P.O.T.W., J.O., D.P., P.B.S., N.T., T.U.N., P.V.H., R.V., C. Vernesi and C.E.G. Data acquisition (validation): R.E.S., K.A.F., D.J.C., C.P.E., J.M., K.M.O., G.S., S.H., C.H., S.P.-E., D.R., H.C., K.C., I.-R.M.R., N.V., W.P.G.-C., P.C.A., J.A.L., J.O., P.V.H. and C.E.G. Data acquisition (additional metadata collection): R.E.S., K.A.F., M.W.B., D.J.C., C.P.E., J.M., K.M.O., G.S., V.B., F.E.Z., P.O.T.W. and C.E.G. Data acquisition (cleaning): R.E.S., K.A.F., K.M.O., F.H. and C.E.G. Data analysis: R.E.S., K.A.F., D.J.C., C.P.E., J.M., K.M.O., G.S., D.L.F., B.H. and C.E.G. Data interpretation (figures): R.E.S., K.A.F., C.P.E., K.M.O., S.P.-E., L.D.B., I.-R.M.R., W.P.G.-C., T.H., B.H. and C.E.G. Drafting the manuscript: R.E.S., K.A.F., D.J.C., C.P.E., J.M., K.M.O., G.S., S.H., K.C., R.E., F.E.Z. and C.E.G. All authors reviewed and approved the manuscript.

**Funding** Open access funding provided by the University of Sydney.

**Competing interests** The authors declare no competing interests.

**Additional information**
**Correspondence and requests for materials** should be addressed to Catherine E. Grueber.

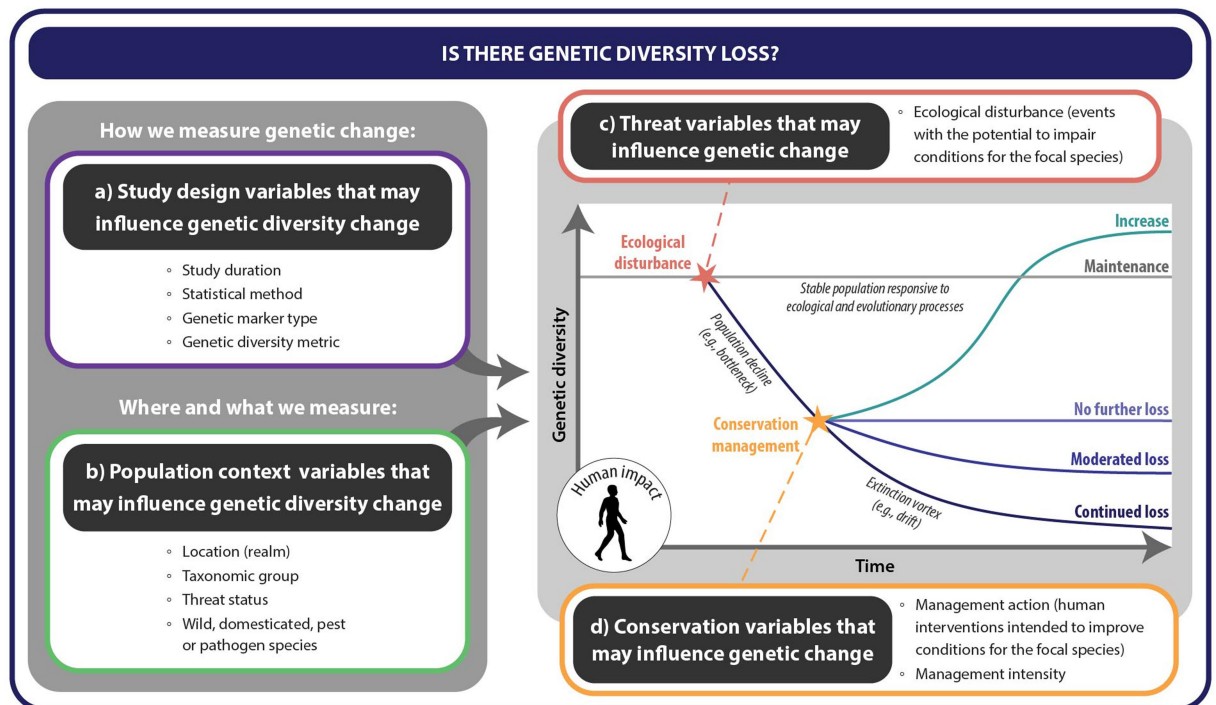

**Extended Data Fig. 1 | Conceptual diagram describing aims and predictions under different scenarios.** Our aim is to determine whether there is overall genetic diversity change, and how this pattern is influenced by: a) study design, b) population context, c) threatening processes, and d) conservation management. When large, stable populations have not encountered or are resilient to disturbance (and thus do not require conservation management), we predict a scenario of '*maintenance*', where no overall genetic diversity change is detected over a timescale relevant to human impact. Alternatively, disturbance may be a negative moderator of genetic change if it results in a loss of genetic diversity (e.g., through a decrease in population size ["bottleneck"]). Following conservation management, we predict three possible outcomes, 1) '*increase*': conservation management reverses loss and genetic diversity returns to initial level (no change detected) or increases beyond initial level (gain detected; management is a positive moderator of genetic diversity change); 2) '*no further loss*' or '*moderated loss*': conservation management halts or mediates loss resulting in a net decline in genetic diversity, but to a lesser degree than if management were absent (i.e., management is a positive moderator of genetic change); or 3) '*continued loss*': conservation management is ineffective or absent, resulting in continued decline (loss detected, e.g., due to genetic drift in small populations). If this trend continues, extinction is inevitable, although the time frame for a given species is hard to predict. Note that the particular timing of an ecological disturbance event or conservation management action in relation to the population's trajectory is not recorded in our dataset, as we focus only on presence or absence of these drivers. In addition to the processes shown, disturbance can also occasionally increase genetic diversity; such data are included in our analysis, but not illustrated here.

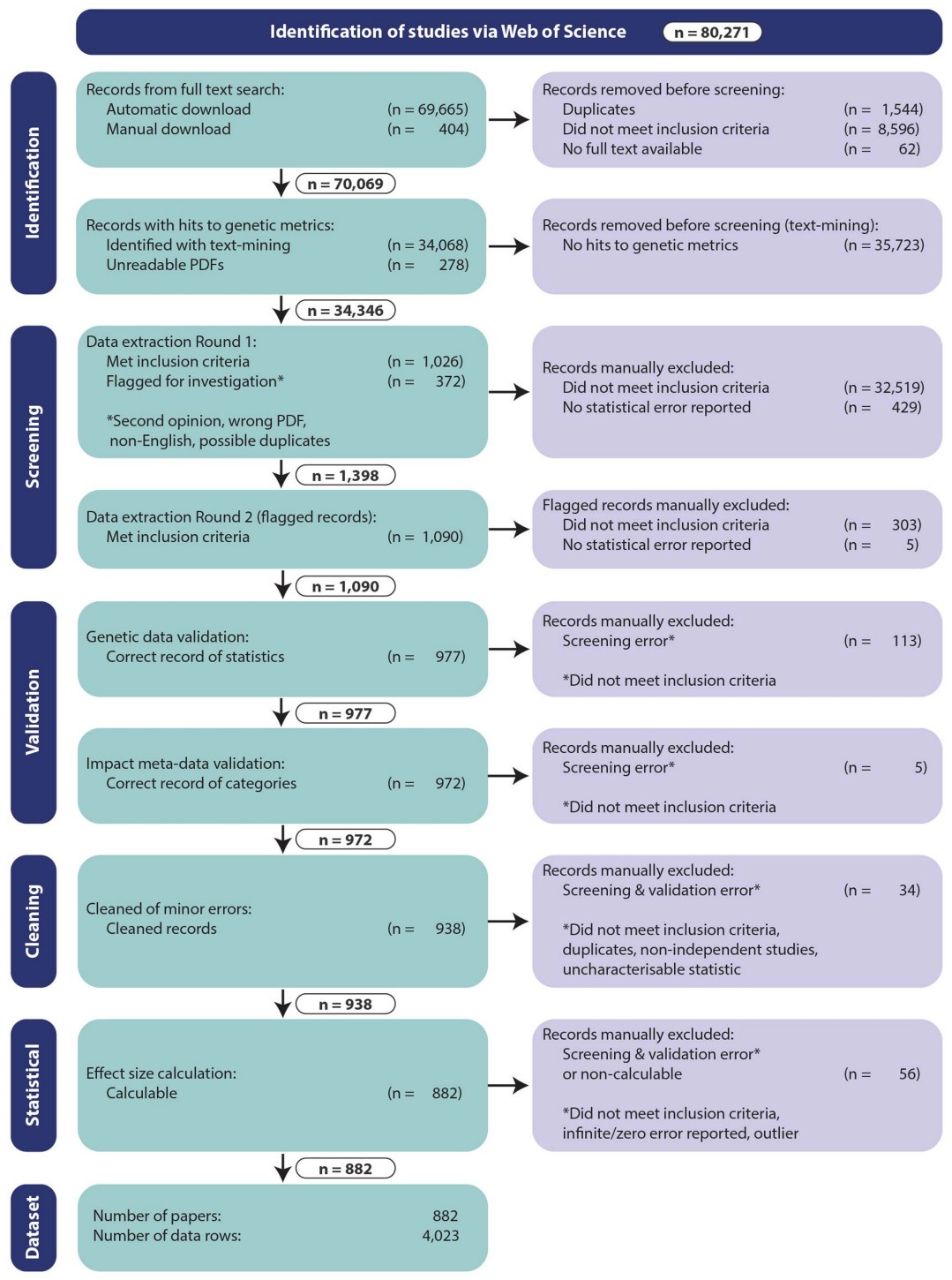

**Identification of studies via Web of Science** n = 80,271

**Identification**

Records from full text search:
Automatic download (n = 69,665)
Manual download (n = 404)

Records removed before screening:
Duplicates (n = 1,544)
Did not meet inclusion criteria (n = 8,596)
No full text available (n = 62)

n = 70,069

Records with hits to genetic metrics:
Identified with text-mining (n = 34,068)
Unreadable PDFs (n = 278)

Records removed before screening (text-mining):
No hits to genetic metrics (n = 35,723)

n = 34,346

**Screening**

Data extraction Round 1:
Met inclusion criteria (n = 1,026)
Flagged for investigation* (n = 372)

*Second opinion, wrong PDF, non-English, possible duplicates

Records manually excluded:
Did not meet inclusion criteria (n = 32,519)
No statistical error reported (n = 429)

n = 1,398

Data extraction Round 2 (flagged records):
Met inclusion criteria (n = 1,090)

Flagged records manually excluded:
Did not meet inclusion criteria (n = 303)
No statistical error reported (n = 5)

n = 1,090

**Validation**

Genetic data validation:
Correct record of statistics (n = 977)

Records manually excluded:
Screening error* (n = 113)

*Did not meet inclusion criteria

n = 977

Impact meta-data validation:
Correct record of categories (n = 972)

Records manually excluded:
Screening error* (n = 5)

*Did not meet inclusion criteria

n = 972

**Cleaning**

Cleaned of minor errors:
Cleaned records (n = 938)

Records manually excluded:
Screening & validation error* (n = 34)

*Did not meet inclusion criteria, duplicates, non-independent studies, uncharacterisable statistic

n = 938

**Statistical**

Effect size calculation:
Calculable (n = 882)

Records manually excluded:
Screening & validation error* or non-calculable (n = 56)

*Did not meet inclusion criteria, infinite/zero error reported, outlier

n = 882

**Dataset**

Number of papers: 882
Number of data rows: 4,023

**Extended Data Fig. 2 | Preferred reporting items for systematic reviews and meta-analyses (PRISMA).** Flow diagram detailing the five steps (identification, screening, validation, cleaning and statistical) taken to generate the systematic review dataset, with sample sizes (n) representing the number of papers (note that a further 40 rows of data from 11 papers, representing extreme values, were removed prior to conducting meta-analysis, see Supporting Information 1.4-1.5).

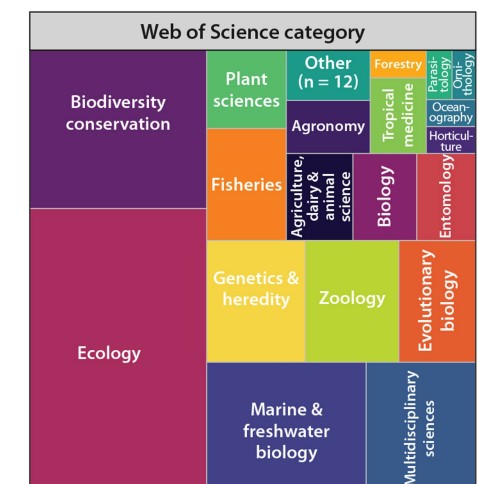

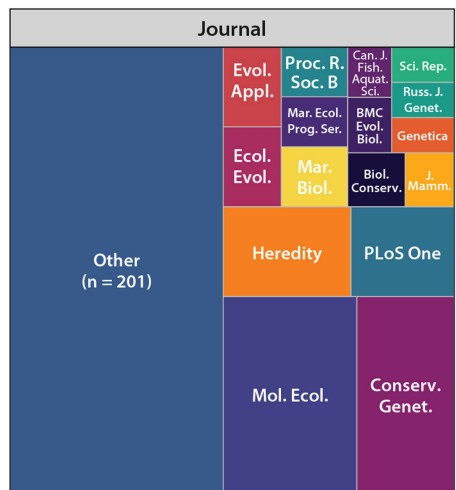

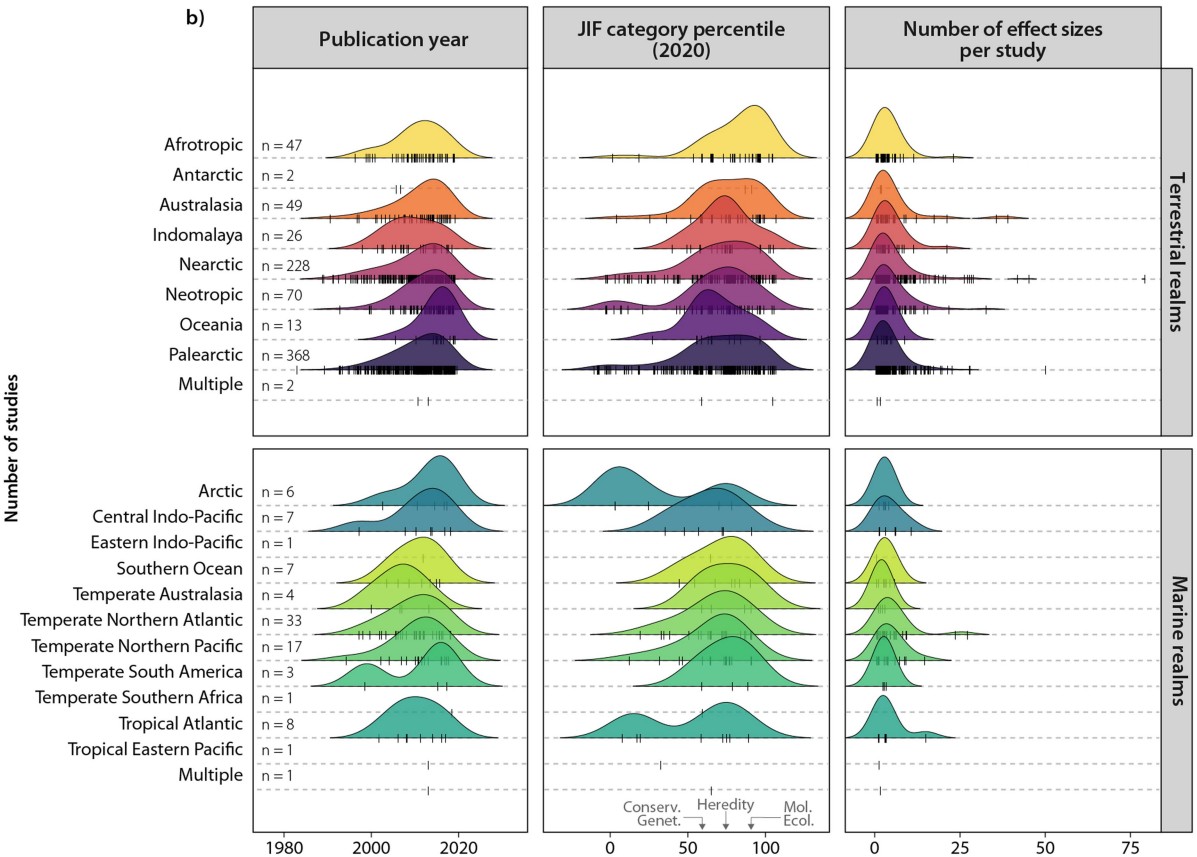

**Extended Data Fig. 3 | Summary of the systematic review dataset: research effort as reflected by publication trends.** a) Tree maps illustrating the number of papers in each Web of Science category and scientific journal (total = 217 journals). The Web of Science category labelled as 'Other' provided fewer than 5 journals (left panel). Journals labelled as 'Other' provided fewer than 10 papers (right panel). b) Density plots and raw data (vertical ticks along x-axis) represent publication year of each paper, the journal impact factor (JIF) category percentile (three well-known biodiversity genetics journals [*Conservation Genetics*, *Heredity* and *Molecular Ecology*] are presented for context on the x-axis, bottom panel), and the number of effect sizes collected per paper, across terrestrial and marine realms (n = total number of papers). Papers reporting on multiple unique populations in different realms are counted multiple times, whereas the category called 'multiple' represents single populations where the distribution spans multiple realms.

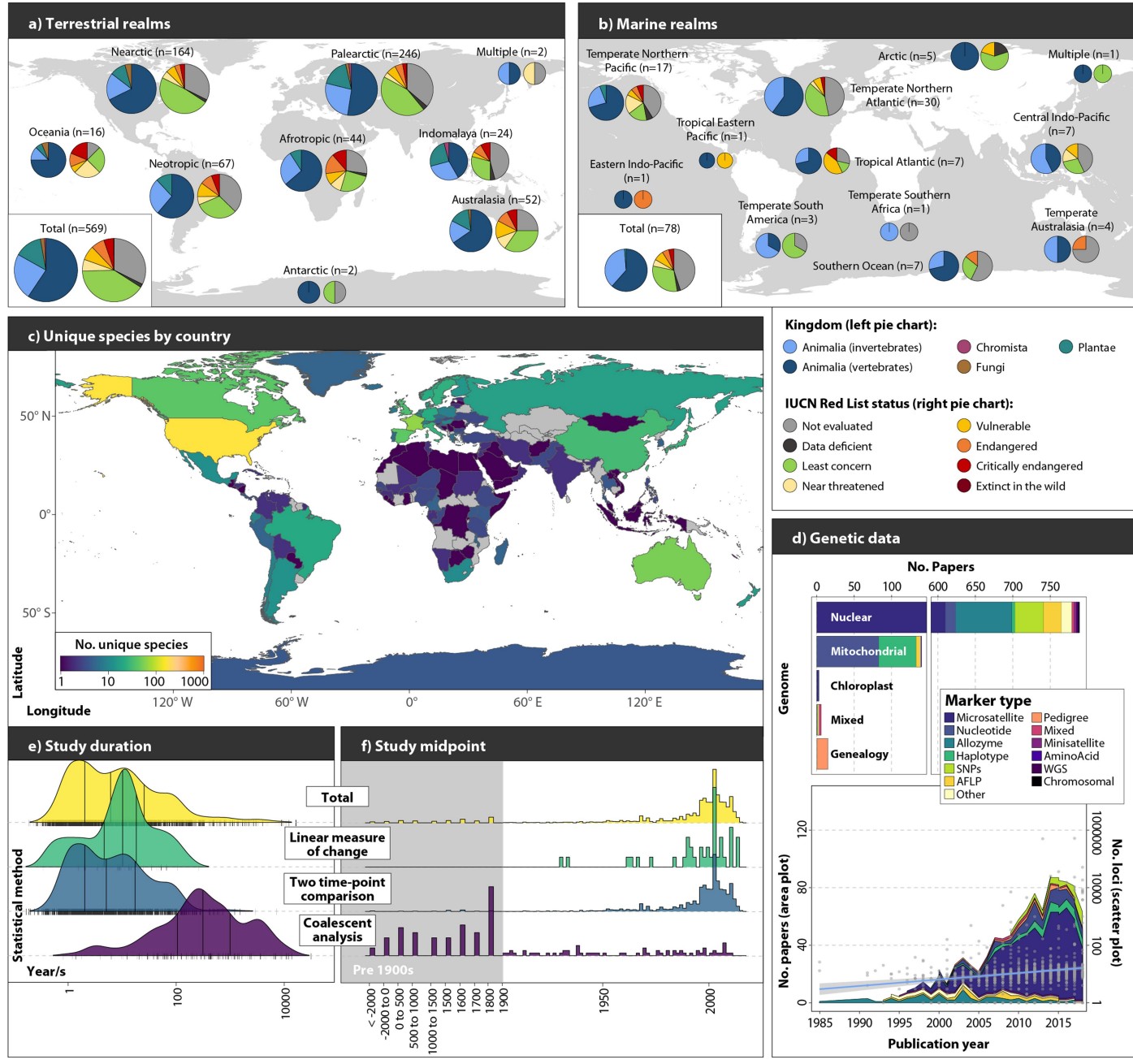

**Extended Data Fig. 4 | Summary of the systematic review dataset: research effort as reflected by study characteristics.** a) The number of unique species (within phyla and IUCN Red List threat status categories) reported from each terrestrial and; b) marine realm (noting that some species occur in both a terrestrial and a marine realm, while those that occur across more than one terrestrial or marine realm are listed as 'multiple'); c) world map where color represents the number of unique species (whereas unique populations are presented in Fig. 1). Gray represents zero counts. Note that both terrestrial and marine realms are represented within the relevant country boundaries, excluding one marine population that could not be reliably linked to a country. Studies spanning country borders are represented multiple times in this figure.

World map modified from ref. 36; d) the number of papers reporting each genetic marker type for each genome (top panel, full definitions in Supporting Information 2.6), and an area plot and scatter plot with a regression line (error band is 95% confidence interval) showing the number of papers reporting different genetic marker types (left axis) and the number of loci reported in papers (right axis) across publication years (bottom panel; noting that one paper published in January 2019 was grouped with the 2018 publications); e) temporal characteristics of study duration (in years) and; f) year midpoint (for the total dataset, and across the three main statistical methods identified in our dataset).

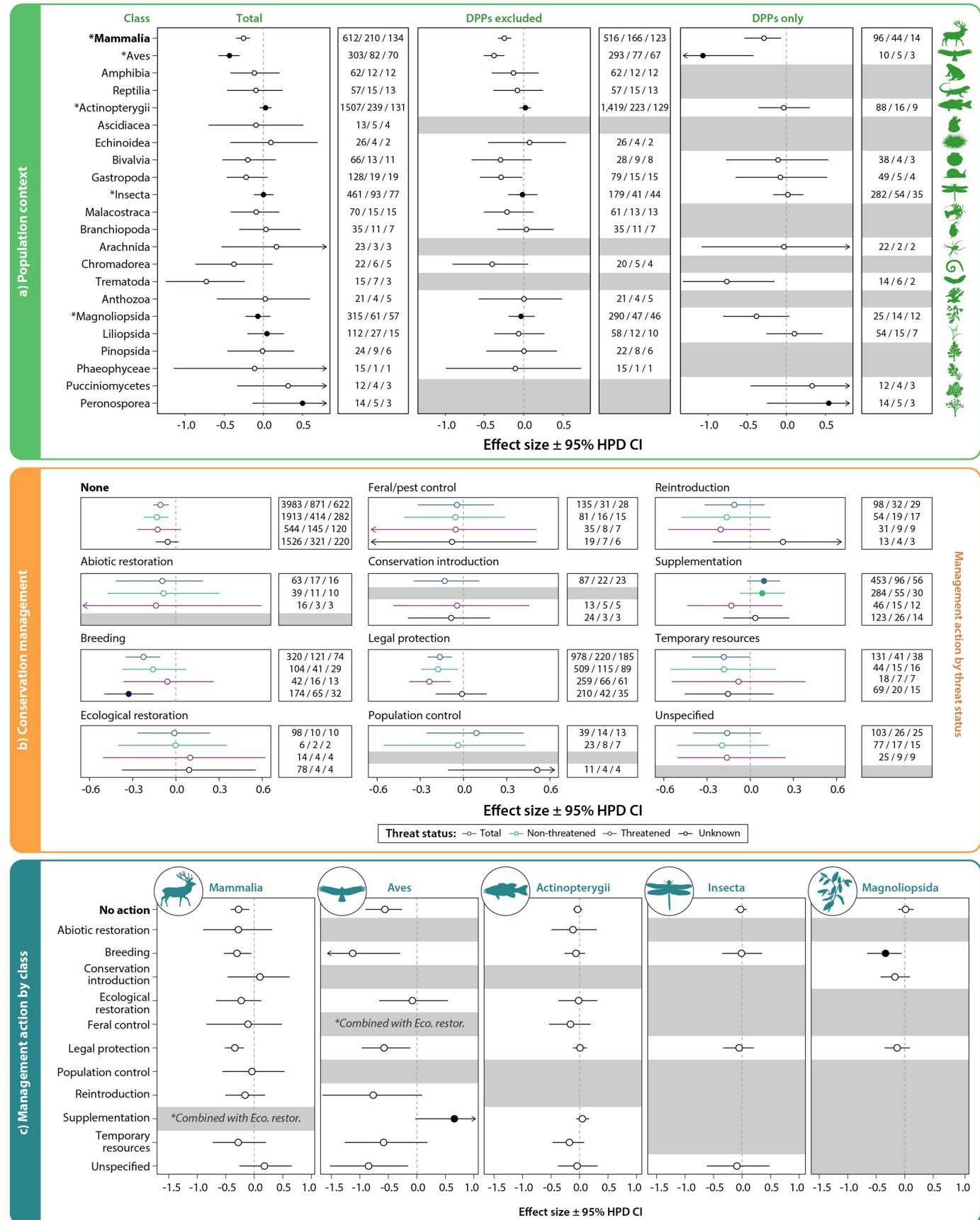

**Extended Data Fig. 5** | See next page for caption.

**Extended Data Fig. 5 | Genetic diversity change across population and conservation contexts.** Meta-regressions (using the reduced meta-analysis dataset) of predicted change in genetic diversity for: a) species in each taxonomic class, for the total dataset (left), excluding domestic, pest or pathogen populations ("DPPs"; middle), and for domestic, pest or pathogen populations only (right), presented in order of phylogeny and including the five most data-rich classes (asterisks) that are also presented in Fig. 2; b) conservation management actions for the total dataset (blue, also presented in Fig. 2); non-threatened species (green); threatened species (light purple); and unknown threat status species (dark purple); c) conservation management action across the five most data-rich taxonomic classes, where asterisks indicate correlated actions that were combined for certain taxa. For all meta-regressions, effect sizes (circles) were measured as mean Hedges' g* and error bars are the 95% highest posterior density credible intervals (HPD CIs). A negative effect size estimate represents a loss of genetic diversity and a positive effect size estimate represents a gain in genetic diversity, statistically significant if the HPD CIs do not overlap zero (dashed line). Arrows denote 95% HPD CIs that extend beyond axis limits. Filled circles represent predictors that are statistically significantly different from the intercept at $\alpha = 0.05$, with the intercept indicated in bold text. Boxes to the right of forest plots provide sample sizes (presented as number of effect sizes / papers / species; see Supporting Information 1.9 for panel c sample sizes). Gray panels indicate variables that were excluded due to insufficient data for modelling (< 10 effect sizes). Estimates for generation and study midpoint (also included as fixed effects) can be found in Supporting Information 1.7 and 1.9. Organism silhouettes obtained from PhyloPic (www.phylopic.org), see Supporting Information 2.6 for image credits.

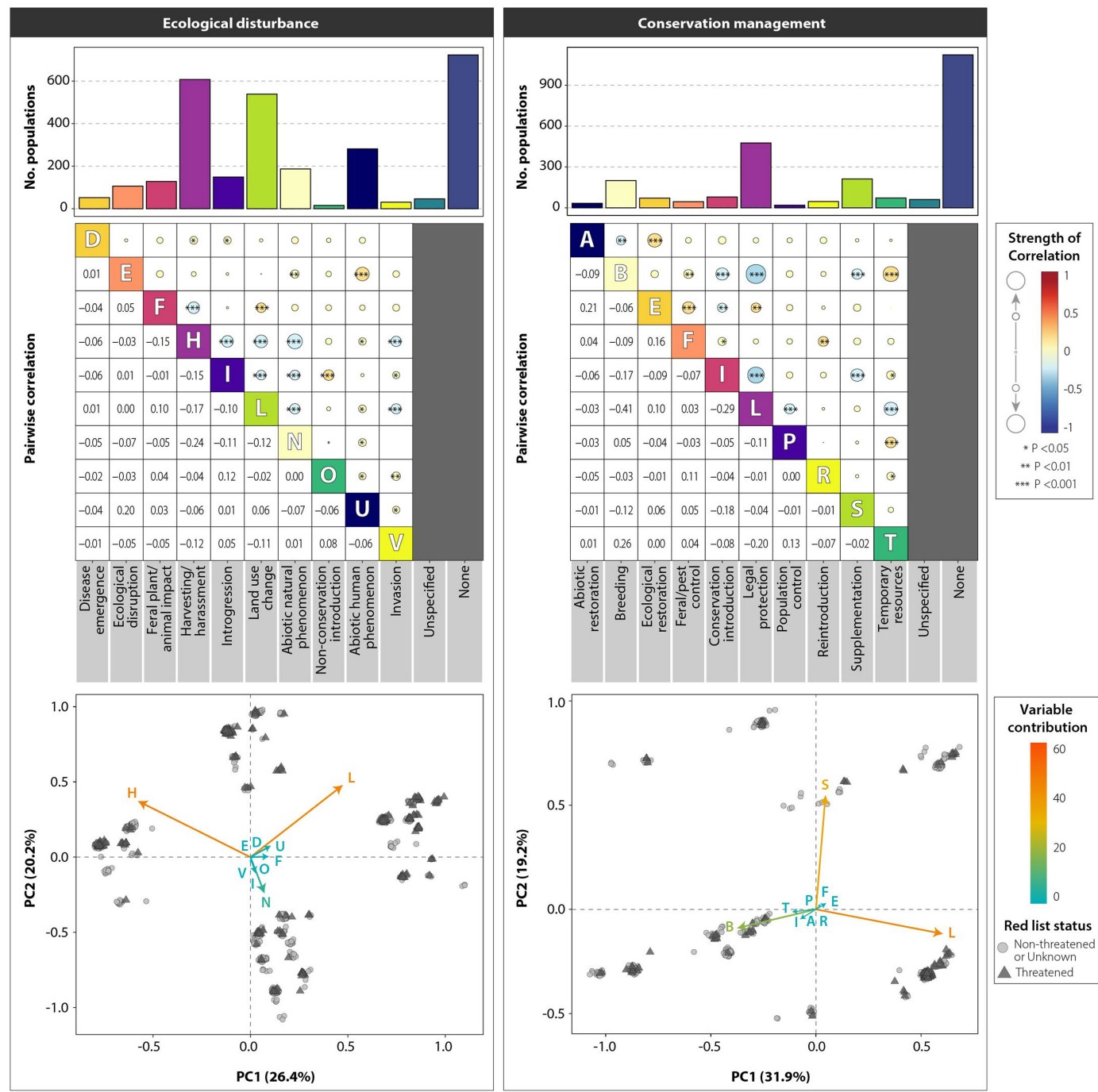

**Extended Data Fig. 6 | Summary of the systematic review dataset; summary statistics for impact meta-data.** Ecological disturbance (left) and conservation management action (right) reported for unique populations of species within each study, including top panel: bar charts of total counts (i.e. sample size) per disturbance or action type; middle panel: correlations between disturbance or action types (lower half = correlation coefficient, upper half = strength of correlation represented by color and size, with asterisks showing statistical significance [two-sided test of $H_0 = 0$, no correction made for multiple testing]); bottom panel: PCA biplots, where letters represent ecological disturbance or conservation management action (as defined in correlation plot above), point colour and shape represent IUCN threat status (light gray circles = non-threatened or unknown, dark gray triangles = threatened), and arrow and letter colour represents the variable contribution to the PCA.

**Ecological disturbance by taxonomic class**

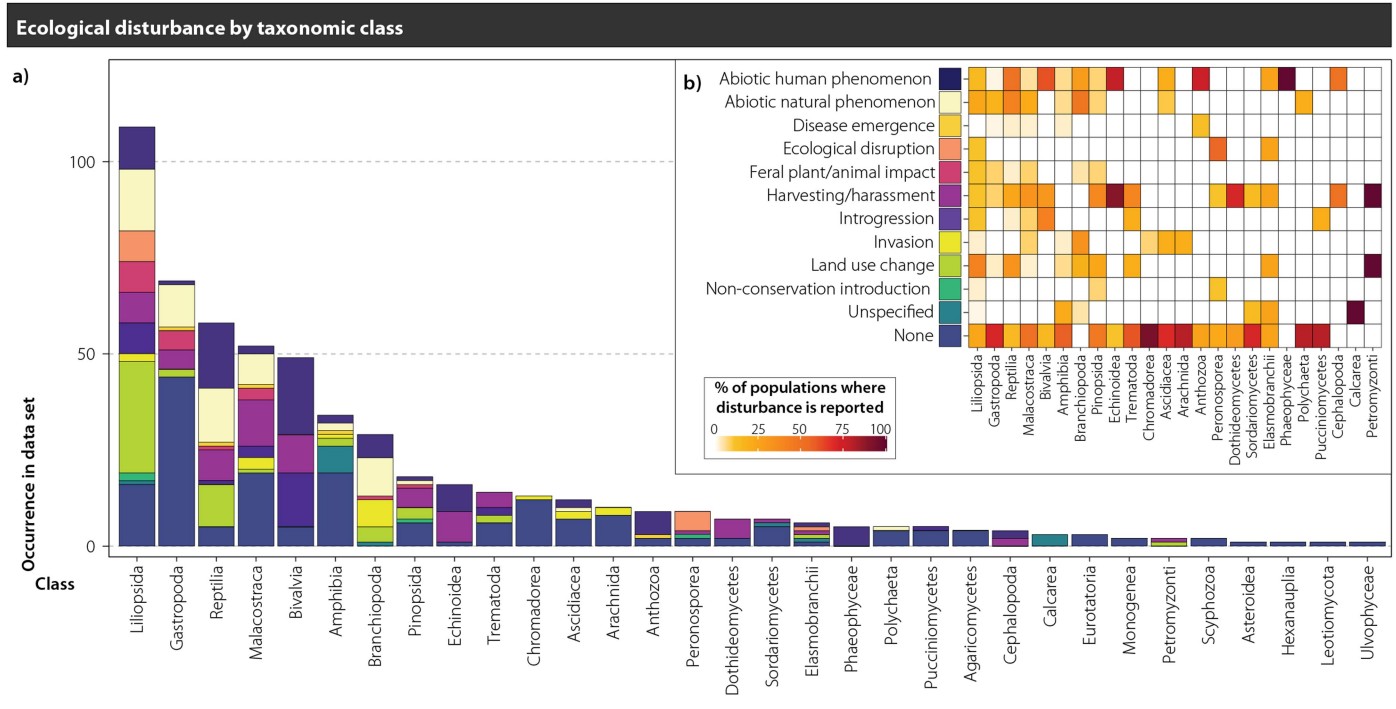

**Conservation management by taxonomic class**

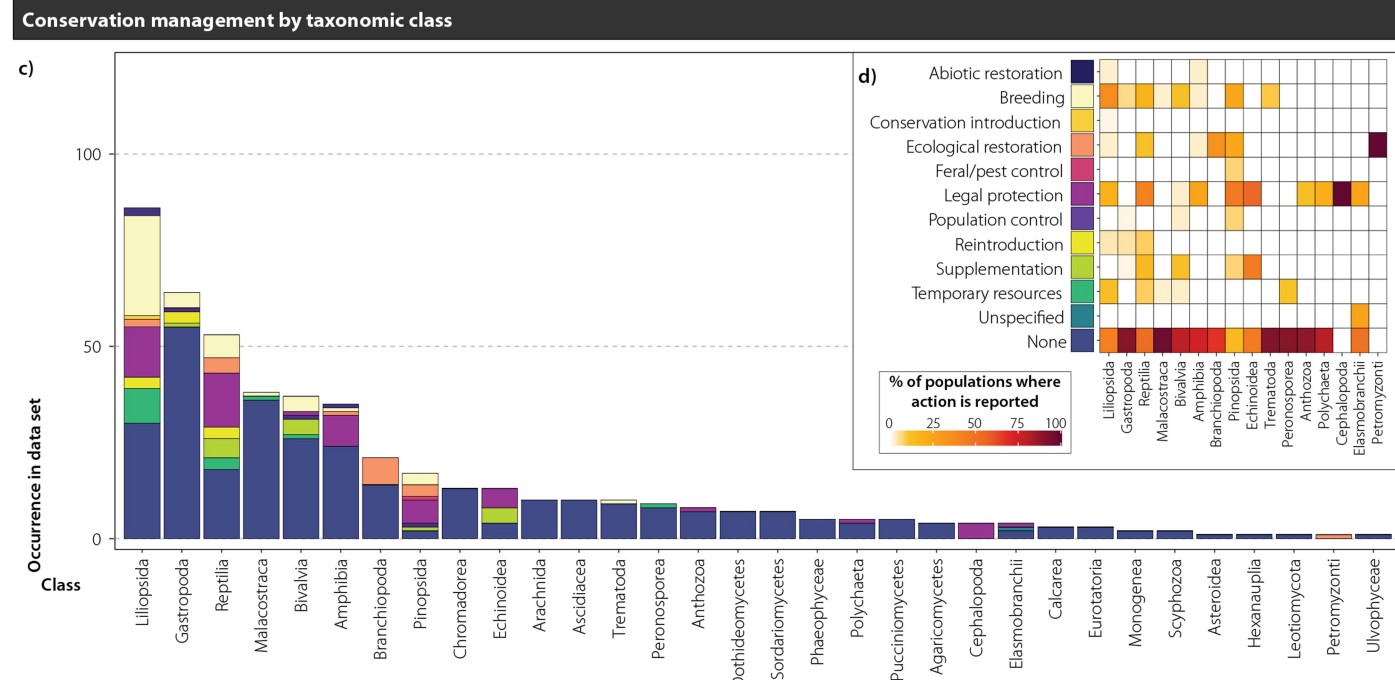

**Extended Data Fig. 7 | Summary of the systematic review dataset; impact meta-data by selected taxonomic classes.** Ecological disturbance (a-b) and conservation management action (c-d) reported for unique populations of species within each study, including stacked bar charts of total counts (i.e., sample size) of the occurrence of each disturbance/action within each taxonomic class (classes are sorted by the amount of data), and heat maps showing the percentage of total unique populations within each taxonomic class for which the different disturbances/actions were reported. The top five most data-rich taxonomic classes (Mammalia, Aves, Actinopterygii, Insecta, Magnoliopsida) are instead presented in Fig. 3, while classes where no action was recorded across all populations are excluded from visualization in heat maps. Colored squares to the right of the disturbance types (b) and conservation actions (d) indicate the color used to represent the disturbances/actions in the bar charts (a, c).

**Extended Data Table 1 | Impact meta-data detailing threatening processes, including ecological disturbance category and brief descriptions (full descriptions provided in Supporting Information 2.5)**

| Category | Description |
|---|---|
| Abiotic human phenomenon | Abiotic processes that result from human actions (e.g., pollution, burning, and processes linked to post-industrial global climate change). |
| Abiotic natural phenomenon | Abiotic processes not directly linked to human activities (e.g., wildfire, flood, river change, geographic change, and pre-industrial climate change). |
| Disease emergence | Emergence of disease in the focal population. |
| Ecological disruption | Ecological change as a result of human activities that impacts the focal population (e.g., loss of prey, nesting cavities or pollinators). |
| Feral animals or plants impacting the study population | The presence of a feral species that impacts the focal species (e.g., weeds, predators, grazers, and competitors). |
| Harvest/ harassment | Killing/stressing the focal species by humans for reasons other than conservation (e.g., hunting, by-catch, poaching, logging, persecution, culling, direct poisoning, and tourism interference). |
| Introgression | Hybridisation as a disturbance, cross-species introgression, and within-species introgression (e.g., unintended hybridisation, and movement of marine organisms in ballast water). |
| Invasion | The focal species is self-introduced to a site that it previously did not inhabit (e.g., range expansions, and colonisation). |
| Land use change | Changes to land use (e.g., clearing, conversion to forestry/agriculture/mining, infrastructure, urbanisation, water abstraction/dams, and generic "habitat fragmentation"). |
| Non-conservation introduction | Accidental or deliberate introduction of a population by humans, but not for the purpose of conserving the introduced species (e.g., biocontrol activities). |
| Unspecified | Ecological disturbance is mentioned or implied, but insufficient details are provided to assign it to a category. |
| None | No ecological disturbances were mentioned in the paper, or disturbance occurred outside of the timeframe relevant to the genetic data reported. |

**Extended Data Table 2 | Impact meta-data detailing conservation management, including management action category and brief descriptions (full descriptions provided in Supporting Information 2.5)**

| Category | Description |
|---|---|
| Abiotic restoration | Activities intended to improve conditions for the focal species via management of abiotic factors (e.g., prescribed burns, controlling river flow volumes, and creating artificial habitat). |
| Breeding | All forms of breeding management (e.g., captive breeding, agriculture, selective breeding, artificial reproductive technologies, contraception, and targeted removal of genetically over-represented breeders). |
| Conservation introduction | Establishing a new site for conservation purposes outside the species' native range. (e.g., establishing new populations of endangered species, introductions for ecological restoration/replanting, and assisted migration). |
| Ecological restoration | Activities that target species other than the focal species, usually (but not always) to improve habitat (e.g., reforestation, aquatic restoration, and re-vegetation). |
| Feral and pest control | Activities that target feral and pest species (both native and non-native), other than the focal species (e.g., removal of competitors, predators, and grazing herbivore). |
| Legal protection | All types of "legal" protection except the action of "IUCN red listing" a species (recorded elsewhere) (e.g., designating a region overlapping the focal population's range as a national or conservation park or reserve, or a restriction on hunting/harvesting). |
| Population control | Controlled harvesting of the focal population to maintain or improve population parameters (e.g., removal of juveniles to improve survival rate). |
| Reintroduction | Focal population established at a site where previously extirpated. |
| Supplementation | Individuals added to an existing population (i.e., often, but not always, for the purposes of genetic and/or demographic rescue). |
| Temporary resources | Provision of short-term resources (not including ecological/abiotic restoration), such as supplementary feeding, nest boxes, fertiliser, and similar. |
| Unspecified | Conservation management is mentioned or implied, but insufficient details are provided to assign it to a category. |
| None | No conservation management actions were mentioned in the paper, including "monitoring only" type studies, or conservation actions were implemented outside of the timeframe relevant to the genetic data reported. |

# Reporting Summary

## Statistics

For all statistical analyses, confirm that the following items are present in the figure legend, table legend, main text, or Methods section.

| n/a | Confirmed | |
|---|---|---|
| ☐ | ☒ | The exact sample size (*n*) for each experimental group/condition, given as a discrete number and unit of measurement |
| ☐ | ☒ | A statement on whether measurements were taken from distinct samples or whether the same sample was measured repeatedly |
| ☐ | ☒ | The statistical test(s) used AND whether they are one- or two-sided<br>*Only common tests should be described solely by name; describe more complex techniques in the Methods section.* |
| ☐ | ☒ | A description of all covariates tested |
| ☐ | ☒ | A description of any assumptions or corrections, such as tests of normality and adjustment for multiple comparisons |
| ☐ | ☒ | A full description of the statistical parameters including central tendency (e.g. means) or other basic estimates (e.g. regression coefficient) AND variation (e.g. standard deviation) or associated estimates of uncertainty (e.g. confidence intervals) |
| ☐ | ☒ | For null hypothesis testing, the test statistic (e.g. *F*, *t*, *r*) with confidence intervals, effect sizes, degrees of freedom and *P* value noted<br>*Give P values as exact values whenever suitable.* |
| ☐ | ☒ | For Bayesian analysis, information on the choice of priors and Markov chain Monte Carlo settings |
| ☐ | ☒ | For hierarchical and complex designs, identification of the appropriate level for tests and full reporting of outcomes |
| ☐ | ☒ | Estimates of effect sizes (e.g. Cohen's *d*, Pearson's *r*), indicating how they were calculated |

*Our web collection on statistics for biologists contains articles on many of the points above.*

## Software and code

Policy information about availability of computer code

| | |
|---|---|
| Data collection | All data used in this study were obtained from published and publicly available sources, as indicated at the "Data" section. |
| Data analysis | Custom text mining code is available on Zenodo DOI: 10.5281/zenodo.13903787 ref#66. Effect size calculation was conducted via established equations provided in Supporting Information 2.11. Meta-analysis was conducted with the MCMCglmm package v 2.34 in R v 3.5.2, with the model equation provided in Supporting Information 2.11. Phylogenetic modelling for sensitivity testing of the base model used phylogenetic relationships established via the Open Tree of Life and the ape package v 5.6.1 in R. We also used the following R-packages: pdfsearch v 0.2.3, dplyr v 0.8.0, stringi v 1.3.1, revtools v 0.4.0, ggplot2 v 3.4.3, treemapify v 2.5.5, ggridges v 0.5.4, corrplot v 0.92, factoextra v 1.0.7, rotl v 3.0.12, ggtree v 3.8.2, ggtreeExtra v 1.10.0, ggimage v 0.3.3, rphylopic v 1.2.1, coda v 0.19.4, and the following software: Zotero v 5.0.60, Endnote v X9 and GetData Graph Digitizer v 2.26. |

For manuscripts utilizing custom algorithms or software that are central to the research but not yet described in published literature, software must be made available to editors and reviewers. We strongly encourage code deposition in a community repository (e.g. GitHub). See the Nature Portfolio guidelines for submitting code & software for further information.

## Data

Policy information about [availability of data](availability of data)

All manuscripts must include a [data availability statement](data availability statement). This statement should provide the following information, where applicable:
- Accession codes, unique identifiers, or web links for publicly available datasets
- A description of any restrictions on data availability
- For clinical datasets or third party data, please ensure that the statement adheres to our [policy](policy)

All datasets associated with this paper are available on Zenodo DOI: 10.5281/zenodo.13903787 [66]. The full bibliography of 882 included papers (including their DOIs) are listed in Supporting Data 1. We used publicly available databases to obtain species characteristics for the 628 species included in our study. Full methods are in Supporting Information 2.6. Generation lengths (see Supporting Data 5) were obtained from scientific literature and databases including Search FishBase (www.fishbase.se/search.php), AmphibiaWeb (www.amphibiaweb.org), CABI Compendium (www.cabidigitallibrary.org/journal/cabicompendium). Threat status was sourced from the IUCN Red List of Threatened Species [30] at June-August 2021. Invasive species status was sourced from the IUCN 100 of the World's Worst Invasive Alien Species list (www.iucngisd.org/gisd/100_worst.php). Pathogen and pest statuses were sourced from the scientific literature and databases including the European and Mediterranean Plant Protection Organization Global Database (gd.eppo.int/), The Global Pest and Disease Database (www.gpdd.info), and CABI Compendium (www.cabi.org/isc).

## Research involving human participants, their data, or biological material

Policy information about studies with [human participants or human data](human participants or human data). See also policy information about [sex, gender (identity/presentation), and sexual orientation](sex, gender (identity/presentation), and sexual orientation) and [race, ethnicity and racism](race, ethnicity and racism).

| | |
|---|---|
| Reporting on sex and gender | N/A |
| Reporting on race, ethnicity, or other socially relevant groupings | N/A |
| Population characteristics | N/A |
| Recruitment | N/A |
| Ethics oversight | N/A |

Note that full information on the approval of the study protocol must also be provided in the manuscript.

# Field-specific reporting

Please select the one below that is the best fit for your research. If you are not sure, read the appropriate sections before making your selection.

☐ Life sciences     ☐ Behavioural & social sciences     ☒ Ecological, evolutionary & environmental sciences

For a reference copy of the document with all sections, see [nature.com/documents/nr-reporting-summary-flat.pdf](nature.com/documents/nr-reporting-summary-flat.pdf)

# Ecological, evolutionary & environmental sciences study design

All studies must disclose on these points even when the disclosure is negative.

| | |
|---|---|
| Study description | A systematic review and meta-analysis of the literature on genetic change over recent (human-impacted) timescales. Measures of genetic change were extracted from the published literature and converted to Hedge's g* effect sizes for meta-analysis (n=4021 effect sizes). Hierarchical MCMCglmm models were fit to account for non-independence as a result of multiple datapoints per study (StudyID random factor), with sensitivity testing to examine the impact of non-independence as a result of phylogenetic relationships between species. |
| Research sample | This study uses data extracted from the published literature. Reporting of the systematic review and meta-analysis follows the global best-practice PRISMA guidelines. Briefly, publications were identified using search strings to query the Web of Science database. Text mining was conducted to refine search results, before manual examination of 34,346 publications. Relevant data were manually extracted as per our study protocol, and data extractions validated by independent authors. A total of 4021 datapoints from 882 publications and 628 species, covering 37 taxonomic classes, were obtained for meta-analysis. All details of the systematic review and meta-analysis are reported either in the Main Article or Supplementary Methods. |
| Sampling strategy | The search string was designed to avoid hypothesis-driven bias around genetic erosion. For example, we included symmetrical search terms such as "gain" and "loss", the text mining related to methods rather than the magnitude or direction of any genetic change, and our manual extraction protocol was agnostic to the directionality of any genetic change (i.e. we reported measures of genetic diversity over time regardless of whether change was observed or not). This resulted in a large dataset of 4021 datapoints. |
| Data collection | After identifying studies meeting our inclusion criteria (Supporting Information 2), we manually extracted data from published |

| | |
|---|---|
| Data collection | records into an Excel template as per the study protocol. A randomly selected subset of 150 papers was independently (and blindly) re-extracted to examine reproducibility. As a result, all data were checked and validated by two teams, one examining the inclusion criteria and genetic data, and the other examining the conservation and ecological disturbance metadata. |
| Timing and spatial scale | The systematic search of the literature was conducted on 18 January 2019. Text mining was conducted shortly thereafter. The manual screening of studies per the inclusion criteria and extraction of data from studies meeting the criteria began at a workshop in Tovetorp, Sweden, in March 2020. Data was collected from papers published between 1985 and 2019 (noting that there was no time constraint on year of publication in the search). The data in this study is not limited by geographic location, and represents the global literature on genetic diversity change. Measures of genetic change span from 10,486 BCE to 2018 CE. |
| Data exclusions | Data were excluded at various steps as reported in the PRISMA flowchart (Extended Figure 1 and Supporting Information 1.1). Data was excluded if it did not meet our pre-specified inclusion criteria, such as by not reporting genetic metrics, if it was duplicated in the dataset, or because statistical measures of error were not reported in the primary study. Additional data was excluded if the statistic was directionless for genetic change (e.g., FST), could not be converted to effect sizes (e.g., datapoints with infinite measures of variance), or as outliers. Sensitivity testing was performed to examine the impact of excluding extreme values. All details of data exclusions are reported in the Supporting Information. |
| Reproducibility | A subset of data extractions were independently and blindly re-extracted by other members of the team to the original extractors. As a result, all data were re-examined and validated by two teams as described in the Supporting Information 2.10 (Repeat extractions and Validation). All Bayesian MCMCglmm meta-analytic models were run in triplicate to calculate a Gelman-Rubin convergence diagnostic of <1.1, ensuring that model results were consistently reproducible and not subject to chain divergence. |
| Randomization | The systematic review dataset after text mining of 34,346 publications was grouped thematically into 16 groups (Supporting Table 2.2a) based on text mining of keywords. Within these groups, studies were randomly split into batches of 100 papers for manual screening, and authors randomly selected a batch within a theme of their knowledge to screen. No further randomization was applicable in this study. |
| Blinding | A subset of extracted data from 150 papers was re-examined by independent members of the team that were blind to the original extractions to examine reproducibility of the study extraction protocol. As a result, more targeted efforts were conducted to validate both the genetic and metadata fields of the entire dataset. |

Did the study involve field work?  ☐ Yes  ☒ No

# Reporting for specific materials, systems and methods

We require information from authors about some types of materials, experimental systems and methods used in many studies. Here, indicate whether each material, system or method listed is relevant to your study. If you are not sure if a list item applies to your research, read the appropriate section before selecting a response.

## Materials & experimental systems

| n/a | Involved in the study |
|---|---|
| ☒ | ☐ Antibodies |
| ☒ | ☐ Eukaryotic cell lines |
| ☒ | ☐ Palaeontology and archaeology |
| ☒ | ☐ Animals and other organisms |
| ☒ | ☐ Clinical data |
| ☒ | ☐ Dual use research of concern |
| ☒ | ☐ Plants |

## Methods

| n/a | Involved in the study |
|---|---|
| ☒ | ☐ ChIP-seq |
| ☒ | ☐ Flow cytometry |
| ☒ | ☐ MRI-based neuroimaging |

## Plants

| | |
|---|---|
| Seed stocks | *Report on the source of all seed stocks or other plant material used. If applicable, state the seed stock centre and catalogue number. If plant specimens were collected from the field, describe the collection location, date and sampling procedures.* |
| Novel plant genotypes | *Describe the methods by which all novel plant genotypes were produced. This includes those generated by transgenic approaches, gene editing, chemical/radiation-based mutagenesis and hybridization. For transgenic lines, describe the transformation method, the number of independent lines analyzed and the generation upon which experiments were performed. For gene-edited lines, describe the editor used, the endogenous sequence targeted for editing, the targeting guide RNA sequence (if applicable) and how the editor was applied.* |
| Authentication | *Describe any authentication procedures for each seed stock used or novel genotype generated. Describe any experiments used to assess the effect of a mutation and, where applicable, how potential secondary effects (e.g. second site T-DNA insertions, mosiacism, off-target gene editing) were examined.* |

