## [Peer Review File · Nature]

Global meta-analysis shows action is needed to halt genetic diversity loss

Corresponding Author: Dr Catherine Grueber

Version 0:

Reviewer comments:

Referee #1

(Remarks to the Author)

I have had the opportunity to review the manuscript titled "Conservation action is needed to halt pervasive genetic diversity loss" submitted for publication in Nature. The paper, using a meta-analysis, proposes a significant contribution towards understanding the loss of intra-specific genetic diversity by shedding light on the factors influencing genetic diversity across various species and conditions. Its potential to interest multiple disciplines (i.e., conservation genetics, conservation biology, population genetics, ecology), including the policy agenda on protecting genetic diversity, is commendable. However, while the study demonstrates considerable promise, there are several areas where improvements are necessary to fully realize its potential and ensure the robustness of its findings. Below, I outline my primary concerns and recommendations.

1. Analysis of Genetic Change Under Different Sets of "Explanatory" Variables:

The authors examine how different contexts (i.e., conservation management) may control genetic diversity across different taxa. I wonder if this can be expanded to also add the geographical location at the continental level, for example. I acknowledge that conducting a meta-regression across different sets of contexts altogether (taxa, population contexts, threats, and interventions) may come with challenges related to model complexity, potential for overfitting, and data requirements. However, conducting a meta-regression across several moderators from different sets of contexts (population context, threats, conservation management) rather than just within each category or across two categories might be indeed possible and can offer a more holistic understanding of the factors influencing genetic change. Instead of treating each category (study design, population context, ecological disturbances, conservation interventions) separately, authors may consider including variables from all these categories in the same model. This might involve using a mixed-effects model with the moderators from different context sets as fixed effects and the studies as random effects, similar to the approach already used in the study.

I wonder also whether it would be possible to fully explore interaction terms between moderators from different categories to explore how the impact of one factor might change depending on the level of another factor. For example, comparisons across different sets of conditions may for example reveal whether intervention of threat impacts change jointly across taxa and regions.

In other words, I miss finding under what conditions the largest declines are observed across the different sets of conditions analyzed. Implementing this suggestion and addressing the associated pros and cons would enhance the manuscript's analytical depth and may offer more specific results when coming to know what taxa, where, and under what threats and interventions the largest declines are happening: what are the pathways to greatly loss genetic diversity. This would help to prioritize regions, taxa, and interventions that may help prevent erosion of genetic diversity.

Moreover, the authors mention that to fit multi-level hierarchical models they use the R package MCMCglmm 2.34. The Bayesian approach incorporated in MCMCglmm allows for the incorporation of prior knowledge into the analyses, offering a comprehensive way to estimate the effects and uncertainties of complex mixed models. A key aspect of Bayesian approaches is the choice of prior. Authors mention that they are using an inverse-gamma prior but do not justify why this choice is the most appropriate. Please discuss this aspect. Also, and even it may be obvious for readers knowing MCMCglmm, it would be adequate to mention that authors are using a Bayesian approach.

2. Spatial Coverage:

I like that the authors present in Figure 2 the spatial coverage at a continental scale. However, I would prefer to also see in

the SI a figure showing the actual location of the studies within continents. I assume that in the reviewed papers, it may sometimes be difficult to pinpoint the specific location of the study, but any effort in this sense would help to have a better grasp of the locations of the planet with a large gap in results. Including a map to illustrate the global distribution of the studies reviewed would provide valuable context for the readers, highlighting the geographical scope and any potential biases in spatial coverage.

3. Clarifications on Hedge's g :

It is my understanding that some of the main assumptions behind Hedge's g are the linearity in effect size and the homogeneity of variances. In other words, the calculation assumes a linear relationship between the measures used and genetic diversity and that the variances within each of the groups being compared are homogeneous. Is this the case? Please discuss. Are there better alternatives out there? Clarifying these choices and discussing their implications would improve the manuscript's transparency and reliability.

4. Generation Time Estimates:

A key aspect to estimate genetic diversity change relates to the estimates used to quantify generation time. Authors use, when possible, the generation time reported in the original papers and explore the literature to fill up gaps in the database. However, this implies that authors are using types of estimates across species that may be difficult to compare. For example, for *grus leucogeranus*, authors use Age at first breeding, but published estimates of modeled generation times is approximately 15 years...see "Generation lengths of the world's birds and their implications for extinction risk" (wiley.com). If I were a future reader, I would wonder how using different proxies to estimate generation time may drive results. I would suggest authors choose 3 to 5 species with short generation times, 3 to 5 species with average generation times, and 3 to 5 species with long generation times and simulate for this subset of species what would be the effect of using different proxies to estimate generation time and how those choices trickle down across analysis.

5. Coalescent Analysis:

The use of coalescent analysis and its potential impact on the results should be critically examined. Presenting the results without the coalescent based-estimates could offer more robust insights.

6. Presentation and Interpretation of Results:

The manuscript appears to present a somewhat inconsistent picture of genetic diversity loss, with potential discrepancies between the abstract and the main text. For example, in line 219 authors state that results reveal a small genetic diversity loss...a loss that is only present in some of the taxa (i.e., in birds but not in insects)...however, in the abstract and other parts of the main text, they somehow oversell that loss, but using the word "persistent"...a more nuanced interpretation of the main findings would be commendable and it would enhance its credibility.

7. Relevance of Sections:

I find the section within the discussion on "Methods for genetic monitoring" to be of lesser relevance when discussing the main aspects of the study. The idea that more genomic data would improve our knowledge of global genetic diversity has been repeated across the literature a plethora of times. Streamlining content to focus on key findings and their implications would enhance clarity and impact. I recommend revisiting the manuscript structure to remove or revise this section. I would remove this section and use that space to come with policy recommendation about what, where and how we should be protecting. These more oriented practical recommendations seems to be lacking from the manuscript.

8.- On Insects. Although the magnitude and characteristics of insect population declines are still a matter of debate in the scientific community, there is an overall agreement on the fact that any species of insects are in declines. However, the results here show no genetic decline in insects. Can authors discuss why this apparent contradiction?

Other Small Comments:

A) Line 258- I cannot find the r 0.41 in the extended figure 5.

B) Line 279- The wording in this context can be understood as significant from a statistical point of view...change.

C) Line 330 – adjust the order of the impact factors to their actual reported magnitude of impact. Also, the main text should refer here to where in the SI those classes are reported.

D) What are the abiotic human phenomena? I cannot find this information anywhere.

Referee #2

(Remarks to the Author)

This paper consists of a meta-analysis of temporal measures of genetic diversity using a dataset of published estimates from 628 species. The authors find evidence for widespread losses of genetic diversity in their dataset, though the overall magnitude of loss appears to be relatively small. The authors also conduct a detailed investigation of how genetic diversity losses vary by taxonomic/geographic context and are influenced by various threats and conservation actions.

Overall, I found this paper to be compelling and believe it should be of broad interest to the conservation genetics community. Though I don't have much expertise in meta-analysis to provide detailed comments on the methods, the choices that the authors make in analyzing their dataset generally seem justified. Finally, I found the paper itself to be clear and well-written, and I commend the authors for undergoing the substantial task of creating this dataset and presenting this meta-

analysis with a clear and timely message.

All of that said, my one major concern is that the analysis here is not especially novel and the results are somewhat expected. Specifically, the results here are highly similar to Leigh et al. 2019 *Evol. App.*, which also reported a relatively small but significant decline in genetic diversity when conducting a meta-analysis based on data from 91 species with somewhat similar taxonomic composition. Although the scope and detail of this present study is much larger, the qualitative findings are very similar. Thus, I am not 100% certain whether this analysis meets the high bar of novelty often expected of Nature papers.

Below are some more minor comments:

Line 219: How are we meant to interpret these values in the absence of units? Some explanation could be helpful.

Line 265: I am confused how there appear to be so many instances of "supplementation" in your dataset given that other authors (Fitzpatrick, Mittan-Moreau, Miller, & Judson, 2023; Frankham, 2015) have reported only ~20 or so documented cases of assisted migration for the purpose of genetic rescue? By contrast, Table S1.9 reports 56 species where 'supplementation' occurred. Given that these results are highlighted in the Abstract and Discussion, I think some clarification may be warranted.

Line 342: I may have missed it, but do you report your findings when excluding these domestic, pet, and pathogen populations? I'm not entirely convinced it makes sense to include them in this analysis given that their circumstances are fundamentally different from wild plant & animal populations and they are not subject to the same sort of conservation needs.

Line 479: Can you explain better what you mean by "coalescent analysis" here? This terminology is very vague in this context.

Referee #3

(Remarks to the Author)

The manuscript by Shaw et al. uses a dataset on temporal variation in genetic diversity from 628 species to study trends in genetic diversity over human-impacted time scales and the different factors that cause genetic diversity loss with a particular focus on human impact. In order to gather these data, the authors have made an impressive effort in screening the literature for suitable datasets. Although their final dataset does not include a large number of species, and despite some biases that the authors acknowledge and account for, these taxa cover several major groups across the tree of life and are distributed in diverse marine and terrestrial ecosystems.

After data mining and quality screening the authors use the final dataset to perform meta-analyses where they evaluate the effect of different human activities such as land use change, harassment, harvesting etc and natural phenomena on genetic diversity over time. The authors also test the effect of different conservation actions aimed at restoring biodiversity that are expected to also have effects on genetic diversity. Finally, the authors also consider the potential effect of shared evolutionary history on genetic diversity trends by evaluating the effect of phylogeny on the observed genetic diversity. The authors find that most of the populations were impacted by different threats and that genetic diversity loss is widespread and significant across taxa and ecosystems. However, they also found that less than half of the studied species were subject to conservation actions. The main factors associated with genetic diversity loss were land-use change, disease, abiotic natural phenomena, harvesting, and harassment highlighting the magnitude of human impact. In the cases where some conservation actions were taken, only supplementation was found to have a significant and positive impact on genetic diversity.

Overall, I find the manuscript interesting, and I am impressed by the efforts that the authors have done to collect the data. To my knowledge this is the first study that evaluates temporal trends in intraspecific genetic diversity at such spatial and taxonomic scale. The findings are interesting and very relevant given the current focus on genetic diversity loss and its significance. I think that the authors have done a good job in filtering the studies that they have included in the final meta-analyses. As far as I can evaluate (note that I am not a statistician) statistical analyses seem sound and interpretations are supported by the results. I also like the fact that the authors have chosen a Bayesian framework for their meta-analyses.

In terms of the test for phylogenetic signal and presenting a tree that visualizes the relationships of the study taxa, the authors have used the open tree of life, which given the nature of the dataset and the taxonomic scale of the analyses, is an appropriate strategy. The authors have used the tree they generated to also account for phylogeny in their statistical analyses.

References are appropriate and I did not identify obvious omissions both in the main text and in the various supplementary materials provided.

In conclusion, I find the ms worth of publication after some potential revisions. Here are a few suggestions for potential improvements to the ms shall it be finally accepted for publication.

Although, I think that the approach to phylogeny here is appropriate, I feel that some additional information should be provided. For example, I assume that the authors have extracted a tree with branch lengths in millions of years (dated tree) but this is not specified. Since it is also possible to extract just a topology without branch lengths from the open tree of life, I think it is worth specifying that you have used a dated tree here. Also, in lines 632-633 you mention that you calculated

phylogenetic variance, but you do not specify how you have done this. Please add this information here including what package you used to do that. Same for lambda, what package you used to calculate lambda?

I also found some parts of the ms hard to read (e.g. parts of the results section). Perhaps the flow can be improved a bit although I understand that that may be difficult as there are many results that are presented here and space is limited.

I also have a question regarding your Figure 1. I guess that there is a possibility where initial human impact (ecological disturbance) can lead to increase of genetic diversity. For example, by allowing migration and admixture of previously isolated populations over time. It is perhaps worth to mention that this scenario is not shown here or that migration is not considered.

Referee #4

(Remarks to the Author)

(1) This study involved a huge amount of work to find, extract, analyze, and interpret data on changes in genetic diversity over time. There have been previous syntheses and meta-analyses of genetic diversity changes, but as far as I know, none as comprehensive as this one. It is a very impressive effort. The results appear coherent with earlier analyses, with an average decline over time of fairly small magnitude, although it is somewhat unclear how robust the conclusions are to alternative, and reasonable, analysis decisions.

(2) In terms of originality, I fully agree with the authors first argument about this being the most comprehensive analysis to date. A second point of the originality argument was that the analysis was done “alongside information about threats and management action”, which is true, although it’s not always clear what conclusions can be drawn from this. For example, is a conservation action expected to increase genetic diversity (or at least slow loss), or is it a sign that a population is declining and therefore likely to be associated with large diversity declines? To their credit, the authors recognize this in the text, but the issue of ambiguous interpretation remains.

(3) An implicit point of originality is the conclusion (in the title) that genetic diversity loss is “pervasive”, in the sense that the result applies to most categories of study design and most geographic regions. However, of the five taxa with enough data to be analyzed as categories, only mammals and birds showed genetic diversity decline on average, while fishes, insects, and plants did not (Fig. 3). The latter three taxa each have more species globally than birds and mammals combined, and for plants and insects the difference is of one or more orders of magnitude. So, while loss might be pervasive for birds and mammals (across space and methods), the data don’t suggest that it’s pervasive across the tree of life.

(4) I suspect that the differences among taxa are at least part of the reason why the analyses incorporating phylogeny had an “adverse effect on parameter estimate precision” (SI 1.5). The authors use this as one justification for having “excluded phylogeny from our subsequent meta-analytic modelling”. But is it not possible that the analysis with phylogeny is actually a better test of how pervasive change is across all species, rather than an analysis in which birds and mammals determine the overall trend? If incorporation of a factor leads to greater uncertainty around a parameter estimate, it can be a sign that what looked like greater certainty was in fact based on a false assumption.

(5) I appreciated the effort to assess the influence of extreme values, especially in light of recent results showing strong contrasts in trends in the Living Planet Index depending on whether extreme values are included or excluded (Leung et al. Nature 2020, 588:267-271). Having identified 40 extreme values, the authors here concluded that only 2 were in error, and the other 38 “represented genuine change”. The outliers had minimal influence on the overall estimate of genetic diversity change, but we don’t know what influence they might have had on the bulk of analyses looking at subsets, which also make up most of the interpretations. In SI 1.5 it says that “inclusion of these data presented interpretation concerns for downstream moderator analyses, especially analysis of subsets, as including these values may lead to ambiguous inferences”. In the case of the LPI, the full story depended on knowing what happens with and without extreme values. Because genetic diversity change is presented for many subgroups (>50 in Fig. 3), automatically some caution is needed when interpreting any one of them alone. If inclusion or exclusion of extreme values reversed some conclusions, the robustness of those conclusions would be in question. I’m not sure we could say whether one or the other was more “ambiguous”. We know the overall amount of change is robust to the choice, but not whether the downstream ones are.

(6) I found the applied implications to be communicated in fairly vague terms. For example, on line 361: “Preventing further loss will require genetic considerations to be incorporated early in biodiversity planning, risk assessment and prioritisation. We recommend incorporating genetic diversity into Red Listing, endangered species legislation, and other national/global biodiversity commitments”. For species individually, status reports (as far as I know) do report on any genetic studies already. If genetic diversity is not being incorporated, how exactly could or should it now be done? I wasn’t sure what the recommendations would mean in real terms, beyond what practitioners already do (e.g., all else equal, prioritize a population with high diversity). Another example is on line 426: “our data show that we have both the theoretical and technical means to prevent further loss if we act now”. How exactly do these specific data show that?

(7) Following on the previous point, in several places, the manuscript argues that genetic diversity is important for population resilience, ecosystem services, etc., which makes it difficult not to revisit the two decades-old questions. This first is whether putatively neutral genetic markers are indeed indicative of such demographic and ecological outcomes. Is there evidence

that a Hedges' g of -0.1 for microsatellites is a cause of population decline? I'm not up to date on the literature on this question, so maybe there is; but maybe there isn't. The second question is whether genetic diversity decline is a cause of such ecological outcomes at all, or more often a symptom. Either way, change might be an indicator of something important, but it's unclear how well supported the implication is that it is a cause.

Some minor points:

(8) I think the authors could more clearly communicate the significance of asking whether an effect size was different from the intercept (and not only from zero). I think I got this in the end, but it wasn't crystal clear (and it is mentioned more than once). It is referred to as "biologically or methodologically meaningful", and an extra sentence explaining how so exactly would be helpful.

(9) Line 239: The word "orthogonal" doesn't seem quite right. A correlation of 0.5 does not indicate orthogonality, although it does indicate that each variable captures some independent information.

(10) Line 459: Criterion 3 is difficult to understand: "The timeframe of the study plausibly took place over human-impacted timescales". Some examples of studies rejected at this stage might help clarify. I looked at the SI, although I will admit to not having read every word of the 72 single-spaced pages, so perhaps it's in there, but while reading the main text this criterion was the most difficult to grasp.

Version 1:

Reviewer comments:

Referee #1

(Remarks to the Author)

After carefully reading the answers to reviewers and the new version of the main text and the supplementary materials, I think the authors have satisfactorily addressed most of the points raised in the previous round of reviews. I, therefore, appreciate the time and effort the authors took to provide an improved version of their study. I also would like to thank the authors for their work and talent in providing clear figures that convey a lot of information in an easy and attractive way.

However, I still have some minor concerns that I would like the authors to carefully consider:

a) I greatly appreciate the thorough explanation of the adequacy of Hedge's g and the authors' efforts to tailor Hedge's g to the type of data at hand. I am convinced that this test may be the most appropriate option. However, I am concerned about knowing to what degree the data used fulfills the assumptions of linearity in effect size and homogeneity of variances. I apologize for not being more explicit in my previous review, but I do think the authors should clearly report what portion, or which subsets of data met the assumptions and which did not to enhance the transparency of the analytical protocols, as well as the credibility and reliability of the findings. I am afraid that failing to do so may leave future readers with the impression of not knowing to what degree the test is adequate for the data available.

b) I still find it difficult to interpret the results of genetic loss as "global"...they may be global for mammals and birds but not for the other groups. I suggest the authors use the word "loss" only when referring to the groups that have actually lost genetic diversity. However, I agree with the authors on the likely scenario that, in the coming decades, the loss of genetic diversity in other groups globally will become more apparent. This suggests that we may be on the brink of a global genetic erosion phenomenon, especially given that significant genetic change is challenging to detect when the median study duration is only six years. I know this comment is a matter of details, but sometimes the devil is in the details.

c) In an answer to reviewer 4, the authors wrote, "We do not argue that neutral (e.g., microsatellite) genetic diversity loss is a cause of decline. We argue that decline is a cause of genetic diversity loss, and that genetic diversity loss - even neutral diversity - is therefore cause for alarm." I still find this answer quite difficult to follow in terms of its internal logic. The original statement's logic suggests that population decline causes genetic diversity loss and emphasizes concern about this loss. However, it could be perceived as contradictory. In my view, there is a subtle inconsistency. The authors first separate the impact of neutral genetic diversity loss from being a cause of population decline, but then suggest it is alarming. This could be interpreted as contradictory. Or am I misinterpreting the authors here?

Other minor comments can be found here:

1) Why does the first reference, line 127, start with a 7?

2) Meta-analysis is first mentioned in line 168. I know the authors may be facing limitations on the number of references they can include, but somehow a reference to a paper on the philosophy and methods for meta-analysis may be useful for readers.

3) In line 177, it should read "Using meta-regressions" instead of "Using meta-regression".

4) I am not sure of the specific guidelines in Nature about including p -values when reporting r , but in line 239, r values are not accompanied by the p -values.

5) The header "Disturbance is more common than conservation management" does not fully summarize the main topic or

result of the section. More common in what? I assume that is in reporting, but it is not self-explanatory. Please rephrase for clarity.

6) In lines 364-365, authors report that “losses of genetic diversity...indicates that the population size declines at the basis of these genetic diversity losses must be considerable”. The use of “must” here feels a bit strong. I agree with reviewers that in many cases, population size and genetic diversity correlate, but this is not always true. I think the verb “may” would be more appropriate.

7) In line 527, for improving clarity, can the authors write that studies were suitable for inclusion in our analysis only if they satisfied ALL the following criteria?

8) In figure 1, I find it difficult to relate “How we measure change” and “Where and What we measure” with the text in the boxes below those headers...I would expect to find answers to those questions (i.e., what taxonomic groups, realms, and metrics are part of the study)

9) In Figure 2g, Abiotic Phenomenon and Human Natural are not aligned with a specific row...this is confusing. Please clarify and redraw if needed.

10) In Figure 1g and in Extended Figure 6, there are blue colors that are too similar to each other, making it difficult to understand the figures. Please improve color legends for easier understanding of the figures.

Referee #2

(Remarks to the Author)

I am satisfied with the authors' revision and have no further comments.

Referee #3

(Remarks to the Author)

I have had the opportunity to review an earlier version of the manuscript titled "Conservation action is needed to halt pervasive genetic diversity loss". In this ms the authors perform a meta-analysis where they evaluate the effect of different human activities and natural phenomena on genetic diversity over time. They also test the effect of different conservation actions on genetic diversity. The authors also incorporate phylogenetic data to account for the potential effect of shared evolutionary history on genetic diversity.

The results of the present analyses suggest that that most of the populations are experiencing loss of genetic diversity and that that pattern is widespread and significant across taxa and ecosystems. The authors also show that less than half of the studied species were subject to conservation actions despite pervasive genetic diversity loss. Genetic diversity loss is mainly driven by land-use change, disease, abiotic natural phenomena, harvesting, and harassment. The authors find that only supplementation had a significant and positive impact on genetic diversity.

In this reviewed version of the manuscript the authors have made a great effort to address my comments and those of additional 3 reviewers. In my opinion they have carried out a thorough revision and they have address satisfactory all points that I have raised in my comments.

At that stage I do not have any further comments and I am happy to recommend the manuscript for publication.

Referee #4

(Remarks to the Author)

The authors have provided a thorough and thoughtful response to all of the critiques and points of confusion presented by the reviewers. I'm not qualified to comment on all of the technical points of analysis, but the authors appear to have considered each criticism carefully, with responses that strike a balance between working with the limitations of the data and extracting from them as much as possible. This large, comprehensive, and thorough study will be of interest to many people in the field.

I still struggle to fully appreciate the real-world implications of the results, and swapping in “global” for “pervasive” seems mostly cosmetic, but I can see how there is room for different views here. The authors clarify that they can't point causal arrows from disturbances or conservation action to genetic diversity change (multiple, very different interpretation are plausible; point 2 in initial review), and that they basically see genetic diversity decline as an indication of population decline rather than the reverse (point 7). The latter admission would seem clash with statements at the very core of the study, such as “Mitigating genetic diversity loss is a major global challenge” or that we need to “halt genetic diversity loss”, given that genetic diversity is mostly a proxy for the real quantity of interest, population size, on which there is lots of current study and debate.

Version 2:

Reviewer comments:

Referee #1

(Remarks to the Author)

Dear Dr. Catherine E. Grueber and co-authors,

Many congrats for this fantastic effort and piece of research. I also appreciate your willingness to enhance the paper based on the reviewer's recommendations. I don't have any further comments or suggestions, aside from a few minor typos I noticed (see below). I'm looking forward to seeing this paper published.

Line 170 "published records" ..., maybe better "published papers"? Or "studies"?

Line 175 - "across a range of geographic regions, timeframes"...should be time frames

"Our study provides the most comprehensive and impartial investigation of within-population genetic diversity change to date"...what do you mean by impartial? Free of biases??? I would avoid the use of adjectives to self-qualify the robustness of your study...

In extended figure 7...change "% where disturbance reported" by "% where disturbance is reported"

Yours sincerely,
Dr. David Nogués-Bravo

RESPONSE

Referee #1 (Remarks to the Author)

I have had the opportunity to review the manuscript titled "Conservation action is needed to halt pervasive genetic diversity loss" submitted for publication in Nature. The paper, using a meta-analysis, proposes a significant contribution towards understanding the loss of intra-specific genetic diversity by shedding light on the factors influencing genetic diversity across various species and conditions. Its potential to interest multiple disciplines (i.e., conservation genetics, conservation biology, population genetics, ecology), including the policy agenda on protecting genetic diversity, is commendable. However, while the study demonstrates considerable promise, there are several areas where improvements are necessary to fully realize its potential and ensure the robustness of its findings. Below, I outline my primary concerns and recommendations.

1. Analysis of Genetic Change Under Different Sets of "Explanatory" Variables:

The authors examine how different contexts (i.e., conservation management) may control genetic diversity across different taxa. I wonder if this can be expanded to also add the geographical location at the continental level, for example. I acknowledge that conducting a meta-regression across different sets of contexts altogether (taxa, population contexts, threats, and interventions) may come with challenges related to model complexity, potential for overfitting, and data requirements. However, conducting a meta-regression across several moderators from different sets of contexts (population context, threats, conservation management) rather than just within each category or across two categories might be indeed possible and can offer a more holistic understanding of the factors influencing genetic change. Instead of treating each category (study design, population context, ecological disturbances, conservation interventions) separately, authors may consider including variables from all these categories in the same model. This might involve using a mixed-effects model with the moderators from different context sets as fixed effects and the studies as random effects, similar to the approach already used in the study.

AUTHOR RESPONSE: We thank the reviewer for their positive comments on our manuscript and advice on how to improve it. We accounted for geographic location using biogeographic realms (terrestrial or marine) rather than continent, as defined by Dinerstein et al. (2017) and Spalding et al. (2007). The use of realms allowed us to analyse data in biologically meaningful ways (in the sense of distribution and evolution of organisms in these realms), as well as to include marine studies in the same analysis, as marine studies may not otherwise be able to be assigned to a continent (Figure 3b). Furthermore, analyses at the level of realms allowed us to have grouping with enough sample size for analysis. To evaluate results at a smaller geographical scale, we have also added geographic information about the location of each study (by country) in Extended Figure 3.

With regards to the meta-regressions, we set out to collect and examine patterns in genetic diversity change across explanatory variables within the following contexts: 1) study design, 2) population context, 3) threats, and 4) conservation management. Whilst we agree that including variables from all categories in the same model would be ideal, we note both statistical and biological barriers to doing so:

From a statistical perspective, some variables could not be combined due to correlations. For example, unlike all other study design models, the study duration meta-regression did not include the random effect of Year Midpoint as it was correlated with study duration (i.e. studies with a more recent midpoint were of shorter duration). The study design and population context meta-regressions included categorical variables where the intercept represents the estimate for the reference category (e.g. microsatellites in the genetic marker type model). However, the threats and conservation

management meta-regressions represent a matrix of co-occurring categorical variables, where multiple threats and/or multiple conservation management actions may have been reported for a single data point. The intercept in these models therefore represents the threats/actions set to zero, i.e. not occurring. It would therefore not be possible to combine the two different model types due to the differences in fitting the variables.

From a biological perspective, the aim of the study was to examine genetic diversity change across each of our explanatory variables. Combining multiple categorical variables in a meta-regression would make the interpretation of results challenging. In meta-regression, the coefficients estimate how each subgroup differs from the nominated reference subgroup (represented by the intercept) (Deeks et al., 2023). If all population context variables were combined (threat status, taxonomic class, and realm), the intercept would represent an estimate for a narrow combination of reference categories, for example, studies conducted in non-threatened mammals in the Palearctic. Comparisons of categorical moderator variables to the intercept would be near impossible to interpret, e.g. if genetic diversity loss is greater in Aves, is this as compared to Mammalia, or compared to non-threatened or Palearctic studies? In some cases, combining variables would make the estimate of the Intercept either entirely hypothetical or subject to low precision due to small sample size and the unbalanced nature of our data set (i.e., only 7 datapoints are of short duration, linear measure of change, microsatellite marker, and variant counts metric type).

To justify our approach in the manuscript, we have added the following text to the Methods (L754-759): *“In meta-regression, the coefficients estimate how each category differs from the nominated reference group, represented by the intercept (Deeks et al. 2023). As all moderator variables were categorical, we performed separate meta-regressions for each moderator to avoid the confounding effects of correlations and allow for biologically meaningful interpretation of categorical variables relative to the intercept.”*

Deeks, J. J. et al. (editors). Chapter 10: Analysing data and undertaking meta-analyses. In: Higgins, J. P. T. et al. (editors). *Cochrane Handbook for Systematic Reviews of Interventions* version 6.4 (updated August 2023). Cochrane, 2023. Available from www.training.cochrane.org/handbook.

Dinerstein, E. et al. An ecoregion-based approach to protecting half the terrestrial realm. *BioScience* **67**, 534-545, doi:10.1093/biosci/bix014 (2017).

Spalding, M. D. et al. Marine ecoregions of the world: a bioregionalization of coastal and shelf areas. *BioScience* **57**, 573-583, doi:10.1641/B570707 (2007).

I wonder also whether it would be possible to fully explore interaction terms between moderators from different categories to explore how the impact of one factor might change depending on the level of another factor. For example, comparisons across different sets of conditions may for example reveal whether intervention of threat impacts change jointly across taxa and regions.

In other words, I miss finding under what conditions the largest declines are observed across the different sets of conditions analyzed. Implementing this suggestion and addressing the associated pros and cons would enhance the manuscript's analytical depth and may offer more specific results when coming to know what taxa, where, and under what threats and interventions the largest declines are happening: what are the pathways to greatly loss genetic diversity. This would help to prioritize regions, taxa, and interventions that may help prevent erosion of genetic diversity.

AUTHOR RESPONSE: We thank the reviewer for this thoughtful comment. Although we recognise the importance of identifying the conditions at which large genetic diversity change occurs, the study was designed to investigate factors contributing to genetic diversity change, rather than to best predict the absolute amount of genetic diversity change *per se*. We did not have any specific hypotheses in relation to interactions between various moderator variables. We agree that interaction terms may

reveal the amount of genetic diversity change under specific circumstances, but doing so here would risk overfitting the model rather than providing a synthesis of broad trends in our dataset. Evaluation of the interaction of taxonomic class with realm to identify where the greatest losses occur would not be possible, as sample sizes would be very small and unbalanced in many categories. Given the high heterogeneity in our dataset ($I^2_{total} = 99.22\%$), we have been conservative in our model fitting to avoid the overinterpretation of results that may be influenced by a small number of studies.

While we are unable to explore interaction terms, we have endeavoured to place more emphasis on where/ in which taxa/ under what threats the greatest losses are seen. We have clarified text in the Abstract, Results and Discussion to highlight greater losses in mammals and birds, mean loss across most terrestrial realms, and general background loss regardless of whether disturbances or conservation actions were reported. We also collected new data, by re-screening all 882 papers (2,110 unique populations) and recording the country/ies where the study was located. We now provide a new extended figure (Extended Figure 3) to show the number of unique populations/species for which genetic diversity change estimates are available in each country. This provides an indication of research effort, which can also be used for prioritizing regions. We have added new text to the discussion to also highlight these areas:

L461-464: *“The scientific literature was not uniformly distributed. For example, we find that equatorial and tropical zones of continental Africa and Asia (southern areas) are represented the least in our dataset. This is consistent with a geographic bias in conservation science generally: biodiversity knowledge is replete with bias at taxonomic^{7,45,46} and geographic⁴⁷ levels.”*

Moreover, the authors mention that to fit multi-level hierarchical models they use the R package MCMCglmm 2.34. The Bayesian approach incorporated in MCMCglmm allows for the incorporation of prior knowledge into the analyses, offering a comprehensive way to estimate the effects and uncertainties of complex mixed models. A key aspect of Bayesian approaches is the choice of prior. Authors mention that they are using an inverse-gamma prior but do not justify why this choice is the most appropriate. Please discuss this aspect. Also, and even it may be obvious for readers knowing MCMCglmm, it would be adequate to mention that authors are using a Bayesian approach.

AUTHOR RESPONSE: The inverse-gamma prior was selected following the recommendations of the authors of the R package MCMCglmm, as it is generally weakly informative and is the ‘classical’ and common choice when fitting the ‘animal’ model, as we did for the phylogenetic sensitivity testing (de Villemereuil, 2012; https://devillemereuil.legitux.org/wp-content/uploads/2012/12/tuto_en.pdf). We have added *“weakly-informative inverse-gamma prior”* to both the main text (L761) and to Supporting Information 2.11: Additional meta-analytic procedures. We have amended the main text to *“We fitted multi-level Bayesian hierarchical models in the R package MCMCglmm...”* (L704).

2. Spatial Coverage:

I like that the authors present in Figure 2 the spatial coverage at a continental scale. However, I would prefer to also see in the SI a figure showing the actual location of the studies within continents. I assume that in the reviewed papers, it may sometimes be difficult to pinpoint the specific location of the study, but any effort in this sense would help to have a better grasp of the locations of the planet with a large gap in results. Including a map to illustrate the global distribution of the studies reviewed would provide valuable context for the readers, highlighting the geographical scope and any potential biases in spatial coverage.

AUTHOR RESPONSE: We agree with the reviewer that additional locality information would be useful. As mentioned by the reviewer, it was difficult to determine the actual location of studies within continents, which is why we originally only identified studies to broader biogeographic realms (as well as to provide groupings with large enough sample sizes for modelling). However, to further illustrate

the geographical location of studies we have now collected additional data indicating the country from which samples were collected, and added a new figure (see response above regarding Extended Figure 3).

3. Clarifications on Hedge's g :

It is my understanding that some of the main assumptions behind Hedge's g are the linearity in effect size and the homogeneity of variances. In other words, the calculation assumes a linear relationship between the measures used and genetic diversity and that the variances within each of the groups being compared are homogeneous. Is this the case? Please discuss. Are there better alternatives out there? Clarifying these choices and discussing their implications would improve the manuscript's transparency and reliability.

AUTHOR RESPONSE: We appreciate the opportunity to further justify the use of Hedge's g in our data analyses. A single effect size measure, Hedge's g^* , was chosen here to enable synthesis of all data from diverse sources. Both Cohen's d and Hedge's g are based on a standardised mean difference between two values, in our case the "early" and "recent" timepoints. Hedge's g is preferable to Cohen's d as it weights the pooled standard deviation to account for sample size. Hedge's g has also been shown via simulation to outperform other common effect size measures (Cohen's d and Glass's Δ) when the assumption of homogeneity of variance is violated (Marfo & Okyere, 2019). We also used the J sample size bias corrected Hedge's g^* (rather than Hedge's g) to minimise over-inflation of Hedge's g that can occur with sample sizes < 20 , noting that Hedge's g^* converges to Cohen's d in large sample sizes (Hedges et al., 1985).

We now provide more context in our revised version of the manuscript to justify our choice of effect size measure at L643-647: "*Hedge's g^* was selected as the effect size measure as it is based on the standardised mean difference between two values, in our case the "early" and "recent" timepoints, minimises over-inflation of effect size estimation in studies with sample sizes < 20 , and outperforms other common effect size measures such as Cohen's d and Glass' Δ when the assumption of homogeneity of variance is violated (Marfo & Okyere, 2019).*"

Hedges, L. V., Olkin, I., & Hedges, L. V. (Ed.) (1985). *Statistical methods for meta-analysis*. Academic Press.
Marfo, P. & Okyere, G. A. (2019). The accuracy of effect-size estimates under normals and contaminated normals in meta-analysis. *Heliyon*. 5(6):e01838.

4. Generation Time Estimates:

A key aspect to estimate genetic diversity change relates to the estimates used to quantify generation time. Authors use, when possible, the generation time reported in the original papers and explore the literature to fill up gaps in the database. However, this implies that authors are using types of estimates across species that may be difficult to compare. For example, for *grus leucogeranus*, authors use Age at first breeding, but published estimates of modeled generation times is approximately 15 years...see "Generation lengths of the world's birds and their implications for extinction risk" (wiley.com). If I were a future reader, I would wonder how using different proxies to estimate generation time may drive results. I would suggest authors choose 3 to 5 species with short generation times, 3 to 5 species with average generation times, and 3 to 5 species with long generation times and simulate for this subset of species what would be the effect of using different proxies to estimate generation time and how those choices trickle down across analysis.

AUTHOR RESPONSE: We are in agreement with the reviewer, and recognise that there is variation in generation length methods and data reported for species across the literature. Therefore, we took a strict approach to collecting additional data for inclusion in our meta-analysis, prioritising repeatability over potential minor discrepancies in individual cases. We assumed that the value for generation

length in the original paper was likely to be the most relevant measure for the corresponding genetic diversity measurement reported. For example, in coalescent studies, generation length is used as a variable in the analysis, so it is important to retain the authors' value even though there are alternative and reasonable definitions for generation length. Only where generation length was not reported did we use additional sources, again, following a strict protocol for data collection (Supporting Information 2.6 – Generation length).

We note that our analyses do not aim to "compare" varying generation lengths. Instead, we use generation length (combined with years) to calculate the number of generations passed during the course of a study, and in doing so control for the amount of "time" over which genetic diversity change is recorded. This allows us to scale the magnitude of loss (Hedge's g) by the number of generations, as the latter is not included in the calculation of Hedge's g .

Given the reviewer's valid comment, we investigated the implications of using generations to scale "time" in our analysis by providing an additional sensitivity analysis, wherein simply the length of the study in years is used to control for study duration, rather than the number of generations of the focal species. Controlling for study length in years rather than generations did not influence the overall estimate of genetic diversity change, suggesting that our results are robust to minor variation in generation length estimates (Supporting Information 1.5 – Sensitivity testing; Supporting Figure 1.5a).

5. Coalescent Analysis:

The use of coalescent analysis and its potential impact on the results should be critically examined. Presenting the results without the coalescent based-estimates could offer more robust insights.

AUTHOR RESPONSE: The reviewer has identified that the inclusion of coalescent analyses represents a unique feature of our dataset, but the majority (94.6%) of our meta-analysis dataset involved two time-point comparisons ($n = 3769$ data points) compared to 63 linear measures of change (1.58%) and 151 coalescent data points (3.79%). We used a broad definition of coalescent to describe these latter studies, referring to methods that probabilistically model past effective population size on the basis of a single sample in time, typically using coalescent theory (Kingman 1982) (see also our last response to Reviewer 2).

In response to this reviewer's comment, we point out that we have investigated the impact of the method used to measure genetic diversity change in the study design statistical method meta-regression (results shown at Figure 3b – statistical method). The estimates for all three methods were negative, with the greatest loss estimated in the linear measure of change studies, then coalescent, and then two time-point comparisons. The overall estimate of genetic diversity change (Figure 3a) obtained with the coalescent-based estimate was very similar to the two time-point comparison estimate which formed the bulk of the data, providing support that our overall results are robust to the statistical method used, and that the comparatively small number of coalescent studies had limited leverage on the main findings.

Kingman, J. F. C. (1982). The coalescent. *Stochastic Processes and their Applications* 13(3): 235-248.

6. Presentation and Interpretation of Results:

The manuscript appears to present a somewhat inconsistent picture of genetic diversity loss, with potential discrepancies between the abstract and the main text. For example, in line 219 authors state that results reveal a small genetic diversity loss...a loss that is only present in some of the taxa (i.e., in birds but not in insects)...however, in the abstract and other parts of the main text, they somehow oversell that loss, but using the word "persistent"...a more nuanced interpretation of the main findings would be commendable and it would enhance its credibility.

AUTHOR RESPONSE: We thank the reviewer for pointing out this perceived inconsistency and acknowledge that a more nuanced interpretation is warranted. We have made three main edits to achieve this: 1) we have removed the word ‘pervasive loss’ throughout, instead using ‘global loss’, which we believe more accurately represents our findings; 2) we more clearly delineate between meta-regressions that investigate how genetic diversity change is measured (i.e. study design, for which we found consistent losses), versus those processes that are biological meaningful (population context, threats, conservation management); 3) we provide more detail where no loss was found, for example, in some realms and taxonomic groups.

We note that genetic diversity losses are only detected easily when effective population sizes change drastically. By way of illustration: a sudden bottleneck from $N_e = 1000$ to $N_e = 100$ is only detectable at the 5% threshold after at least 11 generations, per $H_t = H_0(1-1/(2N_e))^t$. In comparison, a change from $N_e = 10000$ to $N_e = 1000$ would require more than 100 generations of drift to be detected at the 5% threshold. In both cases, these are dramatic population size changes of 90%, yet they only represent small genetic losses. The fact that we find significant losses of genetic diversity across (on average) short time spans, across several taxonomic groups (with more trending negative, though non-significant), indicates that the population size changes at the basis of these genetic diversity changes must be considerable.

7. Relevance of Sections:

I find the section within the discussion on “Methods for genetic monitoring” to be of lesser relevance when discussing the main aspects of the study. The idea that more genomic data would improve our knowledge of global genetic diversity has been repeated across the literature a plethora of times. Streamlining content to focus on key findings and their implications would enhance clarity and impact. I recommend revisiting the manuscript structure to remove or revise this section. I would remove this section and use that space to come with policy recommendation about what, where and how we should be protecting. These more oriented practical recommendations seems to be lacking from the manuscript.

AUTHOR RESPONSE: We thank the reviewer for this comment, which we believe is a fair criticism. We have substantially rewritten the Discussion, including sections “*Methods for genetic monitoring exist*” (L449 onward) and “*Protecting global genetic diversity*” (L477 onward), with a focus on policy implications, and in which we provide four recommendations for monitoring and managing genetic diversity. We also more clearly discuss where the greatest losses were found (mammals and birds, although we now discuss caveats with the other groups), and where there are gaps in temporal genetic data (e.g., Extended Figure 3). It is difficult to be prescriptive at this high level of analysis, but we hope the main messages are now clearer.

8.- On Insects. Although the magnitude and characteristics of insect population declines are still a matter of debate in the scientific community, there is an overall agreement on the fact that any species of insects are in declines. However, the results here show no genetic decline in insects. Can authors discuss why this apparent contradiction?

AUTHOR RESPONSE: We thank the reviewer for their observations regarding our findings for insects. As insect populations are typically very large (millions to billions of individuals), widely distributed, and occur in connected metapopulations, the (coalescent) effective size of such populations is typically very large, unless they are species of acute conservation concern. Declines that are important in relative values (e.g., a 90% decline across the entire distribution) might not be noticeable in the genetic data obtained over short time scales: per generation, a population loses $1/(2N_e)$ of its gene diversity. A decline from 10^6 to 10^5 represents a 90% reduction in population size, but this hardly affects gene diversity in the short term: at mutation-drift equilibrium, the nucleotide diversity of a population with $N_e = 10^6$ and $\mu = 10^{-9}$ is expected to be 0.004, approximately. If this population

undergoes a sudden decline to $N_e = 10^5$, the loss of nucleotide diversity after 100 generations would be high unmeasurable (from 0.004000 to 0.003998, 0.05%). Even when N_e was already low (10^4) such a 90% decline (to $N_e = 10^3$) would only yield a moderate loss over 100 generations (5%). This is because the time to reach a new (and much lower) equilibrium is inversely proportional to N_e (in the latter case, the time to halfway equilibrium would already be >1300 generations). Moreover, many insect species included in the study were eurytopic generalists, or even pest species. As a result, we would only expect to detect genetic losses in insect species as a result of extreme bottlenecks. To discuss the lack of genetic diversity loss in insects in our data set, we have added comments to this effect in our Discussion (e.g., L357-365 and L418-423).

Other Small Comments:

A) Line 258- I cannot find the $r = 0.41$ in the extended figure 5.

AUTHOR RESPONSE: The value is present in the middle panel under conservation management, i.e. the correlation matrix when comparing “B” and “L”.

B) Line 279- The wording in this context can be understood as significant from a statistical point of view...change.

AUTHOR RESPONSE: In response to this and other reviewers’ comments, we have substantially edited this section (“*Some conservation actions may mitigate genetic diversity loss*”, L304 onwards) to clarify our findings and improve flow.

C) Line 330 – adjust the order of the impact factors to their actual reported magnitude of impact. Also, the main text should refer here to where in the SI those classes are reported.

AUTHOR RESPONSE: This section of the Discussion (referring to threats most associated with genetic loss, L386 onwards) has been revised to clarify interpretation of our general observations, and as such does not cite specific details in the SI (details are provided in Results).

D) What are the abiotic human phenomena? I cannot find this information anywhere.

AUTHOR RESPONSE: Examples include pollution, burning, processes linked to post-industrial climate change. We have now added two Extended Tables to the main manuscript that provide brief definitions of our categories for ecological disturbance (Extended Table 1) and conservation management action (Extended Table 2). Full descriptions of both are elaborated at Supporting Information 2.5: Impact meta-data.

Referee #2 (Remarks to the Author)

This paper consists of a meta-analysis of temporal measures of genetic diversity using a dataset of published estimates from 628 species. The authors find evidence for widespread losses of genetic diversity in their dataset, though the overall magnitude of loss appears to be relatively small. The authors also conduct a detailed investigation of how genetic diversity losses vary by taxonomic/geographic context and are influenced by various threats and conservation actions.

Overall, I found this paper to be compelling and believe it should be of broad interest to the conservation genetics community. Though I don’t have much expertise in meta-analysis to provide detailed comments on the methods, the choices that the authors make in analyzing their dataset generally seem justified. Finally, I found the paper itself to be clear and well-written, and I commend

the authors for undergoing the substantial task of creating this dataset and presenting this meta-analysis with a clear and timely message.

All of that said, my one major concern is that the analysis here is not especially novel and the results are somewhat expected. Specifically, the results here are highly similar to Leigh et al. 2019 *Evol. App.*, which also reported a relatively small but significant decline in genetic diversity when conducting a meta-analysis based on data from 91 species with somewhat similar taxonomic composition. Although the scope and detail of this present study is much larger, the qualitative findings are very similar. Thus, I am not 100% certain whether this analysis meets the high bar of novelty often expected of Nature papers.

AUTHOR RESPONSE: Thank you for the positive comments on our manuscript and providing comments to improve it. As stated by the other reviewers, our study is the first of its kind at such a wide this spatial and taxonomic scale. For example, we include ~9 times more species, 23 additional taxonomic classes, and ~40 times more data than were included in Leigh et al. 2019. For example, the most data-rich taxonomic class in Leigh et al. 2019 (Actinopterygii) contained 31 values compared to 1,514 in our study. While Leigh et al. 2019 was an informative and important study, providing an estimate of the magnitude of loss, our study is novel in its breadth, as we not only demonstrate that genetic diversity is lost, but we also analyse genetic diversity change alongside reports of ecological disturbance and conservation management.

Beyond simply providing a larger and more comprehensive study, we note important differences in the statistical approach. Leigh et al. 2019 used unweighted averaging of raw values, by applying three paired t-tests to examine whether mean observed heterozygosity, expected heterozygosity, and allelic diversity differed between timepoints. Synthesising unweighted averages is not considered a meta-analysis (nor do the authors claim their study to be), whereas here we have performed a weighted meta-analysis. The advantages of our approach include that all measures were converted to a single effect size measure, allowing for greater synthesis of different study types, and that a weighted approach incorporates the precision of effect size estimates, so that estimates derived from larger studies are given more weight than small, imprecise estimates that may otherwise have undue influence on results. Further, by fitting meta-analytic models that account for non-independence (for example, from multiple data points taken from a single study), we also control for confounding factors. These methodological advantages provide a substantive analytical advance on previous works in this area.

Below are some more minor comments:

Line 219: How are we meant to interpret these values in the absence of units? Some explanation could be helpful.

AUTHOR RESPONSE: We acknowledge that this is challenging to interpret, and have redrafted this paragraph to explain more carefully the interpretation of our analyses and results (L211-217).

Line 265: I am confused how there appear to be so many instances of “supplementation” in your dataset given that other authors (Fitzpatrick, Mittan-Moreau, Miller, & Judson, 2023; Frankham, 2015) have reported only ~20 or so documented cases of assisted migration for the purpose of genetic rescue? By contrast, Table S1.9 reports 56 species where ‘supplementation’ occurred. Given that these results are highlighted in the Abstract and Discussion, I think some clarification may be warranted.

AUTHOR RESPONSE: We have taken a broad definition of “supplementation” and understand the potential confusion here. Our text now clarifies the definition of supplementation used as simply

“adding individuals to an existing population” (L290), and have included our definition in our new Extended Table 2. We have also made it clear when we are specifically referring to genetic rescue, e.g. at L439-445.

Line 342: I may have missed it, but do you report your findings when excluding these domestic, pet, and pathogen populations? I’m not entirely convinced it makes sense to include them in this analysis given that their circumstances are fundamentally different from wild plant & animal populations and they are not subject to the same sort of conservation needs.

AUTHOR RESPONSE: We thank the reviewer for this suggestion. Overall, we deliberately used broad inclusion criteria for our study to consider genetic change across all scenarios, not just those where conservation needs are a concern. We have nevertheless added an additional test for trends in these groups (domestic, pest, pathogen) specifically because those contexts differ from other wild species, and have now added an additional analysis for “all others” that excludes domestic, pest, pathogen populations (see Extended Figure 4)

Line 479: Can you explain better what you mean by “coalescent analysis” here? This terminology is very vague in this context.

AUTHOR RESPONSE: By “coalescent analysis” we are referring to methods that probabilistically model past effective population size on the basis of a single sample in time, typically using coalescent theory (Kingman 1982). These methods allow estimation of the effective size N_e at time t relative to the current effective population size N_e at time 0. Unlike studies measuring change across two points in time, here a point in time in the past was either reported by the authors as showing a marked change, e.g. a bottleneck of size X at time T , as a result of modelling, or it was selected by us as representing an interval relevant in light of historical human influence. As suggested, we have elaborated our definition at L560-568.

Kingman, J. F. C. (1982). The coalescent. *Stochastic Processes and their Applications* 13(3): 235-248.

Referee #3 (Remarks to the Author)

The manuscript by Shaw et al. uses a dataset on temporal variation in genetic diversity from 628 species to study trends in genetic diversity over human-impacted time scales and the different factors that cause genetic diversity loss with a particular focus on human impact. In order to gather these data, the authors have made an impressive effort in screening the literature for suitable datasets. Although their final dataset does not include a large number of species, and despite some biases that the authors acknowledge and account for, these taxa cover several major groups across the tree of life and are distributed in diverse marine and terrestrial ecosystems.

After data mining and quality screening the authors use the final dataset to perform meta-analyses where they evaluate the effect of different human activities such as land use change, harassment, harvesting etc and natural phenomena on genetic diversity over time. The authors also test the effect of different conservation actions aimed at restoring biodiversity that are expected to also have effects on genetic diversity. Finally, the authors also consider the potential effect of shared evolutionary history on genetic diversity trends by evaluating the effect of phylogeny on the observed genetic diversity.

The authors find that most of the populations were impacted by different threats and that genetic diversity loss is widespread and significant across taxa and ecosystems. However, they also found that less than half of the studied species were subject to conservation actions. The main factors associated with genetic diversity loss were land-use change, disease, abiotic natural phenomena, harvesting, and

harassment highlighting the magnitude of human impact. In the cases where some conservation actions were taken, only supplementation was found to have a significant and positive impact on genetic diversity.

Overall, I find the manuscript interesting, and I am impressed by the efforts that the authors have done to collect the data. To my knowledge this is the first study that evaluates temporal trends in intraspecific genetic diversity at such spatial and taxonomic scale. The findings are interesting and very relevant given the current focus on genetic diversity loss and its significance. I think that the authors have done a good job in filtering the studies that they have included in the final meta-analyses. As far as I can evaluate (note that I am not a statistician) statistical analyses seem sound and interpretations are supported by the results. I also like the fact that that the authors have chosen a Bayesian framework for their meta-analyses.

In terms of the test for phylogenetic signal and presenting a tree that visualizes the relationships of the study taxa, the authors have used the open tree of life, which given the nature of the dataset and the taxonomic scale of the analyses, is an appropriate strategy. The authors have used the tree they generated to also account for phylogeny in their statistical analyses.

References are appropriate and I did not identify obvious omissions both in the main text and in the various supplementary materials provided.

In conclusion, I find the ms worth of publication after some potential revisions. Here are a few suggestions for potential improvements to the ms shall it be finally accepted for publication.

AUTHOR RESPONSE: We thank the reviewer for their comments. We have addressed comments relating to the phylogenetic tree below.

Although, I think that the approach to phylogeny here is appropriate, I feel that some additional information should be provided. For example, I assume that the authors have extracted a tree with branch lengths in millions of years (dated tree) but this is not specified. Since it is also possible to extract just a topology without branch lengths from the open tree of life, I think it is worth specifying that you have used a dated tree here. Also, in lines 632-633 you mention that you calculated phylogenetic variance, but you do not specify how you have done this. Please add this information here including what package you used to do that. Same for lambda, what package you used to calculate lambda?

AUTHOR RESPONSE: The reviewer makes a valid point regarding the usefulness of a dated tree, however unfortunately it was not possible to fit a dated tree due to the highly diverse taxa included in our analysis (from Peronosporae to Mammalia), meaning that a unified dated tree combining all taxa across the scale of our analysis does not exist. We instead extracted the topology of the tree from the Open Tree of Life and have now made this clear in the text (L637-639): *“Due to the taxonomic diversity of species in our study, obtaining a dated tree across all species was not possible and so the topology of the tree was used in modelling.”*

For the corresponding variance, the extended heterogeneity statistic partitions total heterogeneity into its components, including phylogenetic variance ($I^2_{\text{phylogeny}}$). This was calculated as described in the cited reference Nakagawa & Santos (2012) and Nakagawa et al. (2023) (both references now cited, L731).

Lambda was calculated as the variance of the random effect of phylogeny divided by the total variance of all random effects (phylogeny + study ID + residual variance). The method is now described at L731-733: *“For the phylogenetic model, we also obtained lambda (phylogenetic signal,*

H^2) as the variance of the random effect of phylogeny divided by the total variance of all random effects (phylogeny, study ID, and residual variance)".

Nakagawa, S. & Santos, E. S. A. Methodological issues and advances in biological meta-analysis. *Evol Ecol* **26**, 1253-1274, doi:10.1007/s10682-012-9555-5 (2012).

Nakagawa, S. et al. Quantitative evidence synthesis: a practical guide on meta-analysis, meta-regression, and publication bias tests for environmental sciences. *Environ Evid* **12**, 8 (2023). doi:10.1186/s13750-023-00301-6.

I also found some parts of the ms hard to read (e.g. parts of the results section). Perhaps the flow can be improved a bit although I understand that that may be difficult as there are many results that are presented here and space is limited.

AUTHOR RESPONSE: In response to this comment and those of the other reviewers, we have redrafted parts of the manuscript (especially Results and Discussion) to improve clarity and flow.

I guess that there is a possibility where initial human impact (ecological disturbance) can lead to increase of genetic diversity. For example, by allowing migration and admixture of previously isolated populations over time. It is perhaps worth to mention that this scenario is not shown here or that migration is not considered.

AUTHOR RESPONSE: We agree that an increase in genetic diversity is a plausible outcome of ecological disturbances, especially via the mechanisms the reviewer cites. For this reason, our data collection strategy included symmetrical terms for genetic diversity change (e.g. both "increase" and "decrease"), and directionality was not a condition of inclusion in the study. Although such data would have been included in our dataset and analysis, we acknowledge that it is absent from our schematic in Figure 1. We have now amended the caption to point this out and to highlight to the reader that increase of genetic diversity due to ecological disturbance might happen under certain scenarios.

Referee #4 (Remarks to the Author)

(1) This study involved a huge amount of work to find, extract, analyze, and interpret data on changes in genetic diversity over time. There have been previous syntheses and meta-analyses of genetic diversity changes, but as far as I know, none as comprehensive as this one. It is a very impressive effort. The results appear coherent with earlier analyses, with an average decline over time of fairly small magnitude, although it is somewhat unclear how robust the conclusions are to alternative, and reasonable, analysis decisions.

AUTHOR RESPONSE: We thank the reviewer for their positive comments. In relation to this reviewer's comment about the general impact of analysis decisions on the robustness of our findings, we reiterate that we have followed well-established PRISMA guidelines for the systematic review and meta-analysis of research data, as primarily developed for the field of medicine (Page et al. 2021), and following further refinements for biological research (Nakagawa & Santos 2012). We conducted multiple sensitivity analyses to test the implications of analytical decision (including additional analyses following the current peer review; Supporting Information 1.5). Furthermore, although our analysis is much larger in scope, the overall patterns we observe are indeed coherent with other studies in related fields (cited in the Introduction and Discussion). Finally, our analytical approach was developed iteratively and collaboratively among the members of the research team, the vast majority of whom contributed to the actual generation of our research data. Technical methods and

interpretation were presented and discussed at multiple workshops of the authorship team as the work proceeded, providing a level of *a priori* rigor and critique in all analytical decisions (e.g. as described at Supporting Information 2.3-2.10). Altogether, these considerations give us confidence that our analytical approach is robust.

Page MJ, McKenzie JE, Bossuyt PM, Boutron I, Hoffmann TC, Mulrow CD, et al. The PRISMA 2020 statement: an updated guideline for reporting systematic reviews. *BMJ* 2021;372:n71. doi: 10.1136/bmj.n71
Nakagawa, S., Santos, E.S.A. Methodological issues and advances in biological meta-analysis. *Evol Ecol* 26, 1253–1274 (2012). <https://doi.org/10.1007/s10682-012-9555-5>

(2) In terms of originality, I fully agree with the authors first argument about this being the most comprehensive analysis to date. A second point of the originality argument was that the analysis was done “alongside information about threats and management action”, which is true, although it’s not always clear what conclusions can be drawn from this. For example, is a conservation action expected to increase genetic diversity (or at least slow loss), or is it a sign that a population is declining and therefore likely to be associated with large diversity declines? To their credit, the authors recognize this in the text, but the issue of ambiguous interpretation remains.

AUTHOR RESPONSE: We are in agreement with the reviewer that our analysis is correlative in the sense that we have identified ecological disturbances and conservation management actions that are associated with greater or lesser genetic diversity change. We have not identified such processes that *cause* genetic change, and have endeavoured to keep this in mind when revising all parts of the manuscript to avoid ambiguity. In addition to identifying cases where genetic diversity declined, remained stable, or increased alongside reports of ecological disturbance or conservation management, we note that an additional insight provided by our analysis is the relative frequencies of various types of ecological disturbance and/or conservation actions, as provided by the systematic review portion of our study. While we cannot comment on causative relationships, our findings provide a broad overview of which types of actions are likely to be working based on the last 30+ years of conservation genetic research. We hope these findings will highlight the importance/relevance of incorporating genetic data in conservation efforts, and will be used as a starting point when considering management options and subsequent causative research.

(3) An implicit point of originality is the conclusion (in the title) that genetic diversity loss is “pervasive”, in the sense that the result applies to most categories of study design and most geographic regions. However, of the five taxa with enough data to be analyzed as categories, only mammals and birds showed genetic diversity decline on average, while fishes, insects, and plants did not (Fig. 3). The latter three taxa each have more species globally than birds and mammals combined, and for plants and insects the difference is of one or more orders of magnitude. So, while loss might be pervasive for birds and mammals (across space and methods), the data don’t suggest that it’s pervasive across the tree of life.

AUTHOR RESPONSE: We thank the reviewer for this valid observation. In response to this comment and comments from other reviewers, we have carefully revised our manuscript in relation to our claims of “pervasive” genetic diversity change. Our data do show that a general pattern of genetic diversity loss is a realistic expectation in the majority of contexts, and have further identified the nuanced view that some taxa show different patterns. We have also elaborated hypotheses for these differences throughout our Discussion (e.g., L353-361)

(4) I suspect that the differences among taxa are at least part of the reason why the analyses incorporating phylogeny had an “adverse effect on parameter estimate precision” (SI 1.5). The authors use this as one justification for having “excluded phylogeny from our subsequent meta-analytic modelling”. But is it not possible that the analysis with phylogeny is actually a better test of

how pervasive change is across all species, rather than an analysis in which birds and mammals determine the overall trend? If incorporation of a factor leads to greater uncertainty around a parameter estimate, it can be a sign that what looked like greater certainty was in fact based on a false assumption.

AUTHOR RESPONSE: We agree that if phylogeny is a useful parameter, it should be included in models regardless of its impacts on parameter estimation. However, in our case, sensitivity testing that included the random effect of phylogeny estimated its effect as being close to zero, and the phylogeny parameter explained very little variance (the random effect of phylogeny contributed 3.79% of overall variance). We therefore interpreted the addition of a phylogenetic variable as adding noise to the model. We have clarified our justification for excluding phylogeny from further modelling, recognising the poor estimation of the negligible effect of phylogeny, rather than the impact on parameter estimate precision as was previously indicated (though this was indeed a consequence) (Supporting Information 1.5 – Sensitivity testing) “Due to the weak contribution of phylogeny to overall variance, we excluded phylogeny from our subsequent meta-analytic modelling”.

We also note and now address a perceived discrepancy in our results - on the one hand, when considering the entirety of the tree of life captured in our dataset, phylogeny did not explain variance in genetic diversity change; on the other hand, we do see some trends across the five major taxonomic classes that we examined more closely (see responses to Reviewer 1). These results are not necessarily inconsistent, indicating that while some groups show greater losses than others, patterns of loss across taxonomic groups are not necessarily correlated with phylogeny *per se*. We acknowledge this pattern in the manuscript at L249-252.

(5) I appreciated the effort to assess the influence of extreme values, especially in light of recent results showing strong contrasts in trends in the Living Planet Index depending on whether extreme values are included or excluded (Leung et al. Nature 2020, 588:267-271). Having identified 40 extreme values, the authors here concluded that only 2 were in error, and the other 38 “represented genuine change”. The outliers had minimal influence on the overall estimate of genetic diversity change, but we don’t know what influence they might have had on the bulk of analyses looking at subsets, which also make up most of the interpretations. In SI 1.5 it says that “inclusion of these data presented interpretation concerns for downstream moderator analyses, especially analysis of subsets, as including these values may lead to ambiguous inferences”. In the case of the LPI, the full story depended on knowing what happens with and without extreme values. Because genetic diversity change is presented for many subgroups (>50 in Fig. 3), automatically some caution is needed when interpreting any one of them alone. If inclusion or exclusion of extreme values reversed some conclusions, the robustness of those conclusions would be in question. I’m not sure we could say whether one or the other was more “ambiguous”. We know the overall amount of change is robust to the choice, but not whether the downstream ones are.

AUTHOR RESPONSE: It is certainly true that extreme or “outlier” values can provide important signals of major dynamics in natural systems, and we fully agree that investigating extremes is important. We tested the effect of including vs excluding these values via sensitivity analysis to investigate our potential outliers carefully. We found that a small portion of our dataset had extreme measures of change. Our goal was to synthesise data on the global measurement of genetic diversity change, and so we therefore omitted the 1% most extreme portion of our dataset from our main analysis (symmetrically from high and low values, so as to not bias our dataset), so that our overall conclusions were representative of the bulk of data; i.e. the remaining 99% of reported estimates. We opted for this approach so that we could consider that model parameter estimates and trends reflect the general tendency of the data, rather than being driven primarily by a small number of cases with extremely high leverage. Many of the extreme values are biologically plausible data and informative of genetic change, and so we also provide detailed, qualitative summaries of those data in light of our

study objectives (Supporting Information 1.4). We have provided greater transparency around these data by directing the reader to them in the Results section (L223-227) and Methods (L702).

(6) I found the applied implications to be communicated in fairly vague terms. For example, on line 361: “Preventing further loss will require genetic considerations to be incorporated early in biodiversity planning, risk assessment and prioritisation. We recommend incorporating genetic diversity into Red Listing, endangered species legislation, and other national/global biodiversity commitments”. For species individually, status reports (as far as I know) do report on any genetic studies already. If genetic diversity is not being incorporated, how exactly could or should it now be done? I wasn’t sure what the recommendations would mean in real terms, beyond what practitioners already do (e.g., all else equal, prioritize a population with high diversity). Another example is on line 426: “our data show that we have both the theoretical and technical means to prevent further loss if we act now”. How exactly do these specific data show that?

AUTHOR RESPONSE: We appreciate this reflection, and have edited part of our Discussion to focus more specifically on the implications of our findings for the management of biodiversity and acknowledge that in some cases genetic studies are reported in threatened species management reports and other grey literature. In particular, we reiterate those conservation actions we found to be associated with maintaining or restoring genetic diversity (i.e., to align with KMGBF targets and goals). We draw closer connections to the new KMGBF, especially indicator A4, which targets the maintenance of genetic diversity in all species (L453). We have also outlined a 4-step hierarchy of how our genetic results should inform management planning and risk assessment going forward (L486 onward).

(7) Following on the previous point, in several places, the manuscript argues that genetic diversity is important for population resilience, ecosystem services, etc., which makes it difficult not to revisit the two decades-old questions. This first is whether putatively neutral genetic markers are indeed indicative of such demographic and ecological outcomes. Is there evidence that a Hedges’ g of -0.1 for microsatellites is a cause of population decline? I’m not up to date on the literature on this question, so maybe there is; but maybe there isn’t. The second question is whether genetic diversity decline is a cause of such ecological outcomes at all, or more often a symptom. Either way, change might be an indicator of something important, but it’s unclear how well supported the implication is that it is a cause.

AUTHOR RESPONSE: Theoretical and empirical evidence links “neutral” genome-wide diversity to population demographics, inbreeding depression and evolutionary potential (reviewed in Kardos et al. 2021, DeWoody et al. 2021). We do not argue that neutral (e.g. microsatellite) genetic diversity loss is a *cause* of decline. We argue that decline is a cause of genetic diversity loss, and that genetic diversity loss - even neutral diversity - is therefore cause for alarm.

Kardos, M., et al. (2021). "The crucial role of genome-wide genetic variation in conservation." *Proceedings of the National Academy of Sciences* 118(48): e2104642118.

DeWoody, J. A. et al. (2021). The long-standing significance of genetic diversity in conservation. *Molecular Ecology*, 30, 4147–4154. <https://doi.org/10.1111/mec.16051>

Some minor points:

(8) I think the authors could more clearly communicate the significance of asking whether an effect size was different from the intercept (and not only from zero). I think I got this in the end, but it wasn’t crystal clear (and it is mentioned more than once). It is referred to as “biologically or methodologically meaningful”, and an extra sentence explaining how so exactly would be helpful.

AUTHOR RESPONSE: Thank you for the opportunity to clarify this interpretation for readers. We now

provide more-detailed narrative interpretation of the intercepts of our models in the Results (e.g. L216-217).

(9) Line 239: The word “orthogonal” doesn’t seem quite right. A correlation of 0.5 does not indicate orthogonality, although it does indicate that each variable captures some independent information.

AUTHOR RESPONSE: Changed as (L238-241): *“Where studies reported multiple genetic diversity metrics, effect sizes were weakly or moderately correlated ($r = 0.25-0.55$) (Supporting Information 1.6), suggesting the four diversity metric types (see Methods) capture somewhat independent information about genetic diversity change.”*

(10) Line 459: Criterion 3 is difficult to understand: “The timeframe of the study plausibly took place over human-impacted timescales”. Some examples of studies rejected at this stage might help clarify. I looked at the SI, although I will admit to not having read every word of the 72 single-spaced pages, so perhaps it’s in there, but while reading the main text this criterion was the most difficult to grasp.

AUTHOR RESPONSE: These details are provided in Supporting Information 2.4. under the subheading *Timeframe of the measurement*, where we state:

“We excluded studies reflecting prehistoric (phylogeographic) changes in the Pleistocene caused by natural long-term processes. We did not define a specific “time limit” for inclusion of coalescent studies: the relevant timeframe was determined case-by-case using the human context as described by the authors (or in the absence of such, 500 years). In general, this means we targeted genetic diversity changes in the last few hundred years and excluded studies on ancient admixture/expansion e.g., in response to prehistoric climate events or events on “geological” timescales. Colonization of landmasses by humans was considered includable, even if thousands of years ago, if relevant to the population under study.”

To improve clarity in the main document, we now add the following to Methods (L535-539):

“The timeframe of the study plausibly took place over human-impacted timescales, regardless of whether the study organism was actually impacted by human activities (in general, we targeted genetic diversity changes in the last few hundred years and excluded studies on ancient admixture/expansion in response to events on “geological” timescales, further detail in Supporting Information 2.4).”

RESPONSE, second revision

Line numbers refer to “clean” version, without tracked changes

Referee #1 (Remarks to the Author)

After carefully reading the answers to reviewers and the new version of the main text and the supplementary materials, I think the authors have satisfactorily addressed most of the points raised in the previous round of reviews. I, therefore, appreciate the time and effort the authors took to provide an improved version of their study. I also would like to thank the authors for their work and talent in providing clear figures that convey a lot of information in an easy and attractive way.

AUTHOR RESPONSE: We thank the reviewer for taking the time to carefully review our manuscript and for their constructive comments.

However, I still have some minor concerns that I would like the authors to carefully consider:

a) I greatly appreciate the thorough explanation of the adequacy of Hedge’s g and the authors’ efforts to tailor Hedge’s g to the type of data at hand. I am convinced that this test may be the most appropriate option. However, I am concerned about knowing to what degree the data used fulfills the assumptions of linearity in effect size and homogeneity of variances. I apologize for not being more explicit in my previous review, but I do think the authors should clearly report what portion, or which subsets of data met the assumptions and which did not to enhance the transparency of the analytical protocols, as well as the credibility and reliability of the findings. I am afraid that failing to do so may leave future readers with the impression of not knowing to what degree the test is adequate for the data available.

AUTHOR RESPONSE:

We thank the reviewer for clarifying their query, and have taken additional steps to investigate the modelling assumptions further. In relation to the homogeneity of variances, we clarify that these queries only apply to our two timepoint ($N = 3769$) and coalescent ($N = 151$) data, as linear measures of change ($N = 63$) already represent intrinsic estimates of effect (such as linear regression), so we must assume that the authors of those original data have chosen appropriate statistical methods for their dataset. We found, that for 95.9% ($N = 3759 / 3920$) of the testable data in our analysis, the difference between the standard deviation of the early and recent estimates was within the range (1:10, 10:1) tested by Marfo & Okyere (2019) in their simulation models. In their study, Marfo & Okyere (2019) found that Hedge’s g was the most appropriate common effect size under these deviations, which we take as justification for our choice that Hedge’s g is a reliable estimator for our dataset.

For the two time-point data, there was a linear relationship between the early and recent errors (Figure 1 below). For the small portion of our dataset where the error deviation was greater ($N = 161$ effect sizes; 4.1% of the combined two time-point and coalescent dataset, 4.0% of the total meta-analytic dataset), two-thirds of these (66.4%) were from coalescent studies, while the remainder were from two time-point studies. Given that two time-point studies are the vast majority of our total meta-analytic dataset ($3769 / 3983 = 94.6\%$), this pattern indicates a high rate of error deviation in the coalescent data (Figure 2, below), compared to the rest of the dataset.

Figure 1: Two time-point comparison studies (N = 3,769) show a linear relationship between early and recent errors (SD = standard deviation)

Figure 2: Coalescent analysis data (N = 151) contain a large number of comparisons where the errors for early and recent measurements of diversity were more different than (1:10, 10:1) (SD = standard deviation).

An error deviation greater than (1:10, 10:1) does not necessarily indicate that our data are biased, and may even be expected under very large changes in mean genetic diversity over time, in line with Taylor’s law (Taylor 1961) and other large-effect analyses (Grissom 2000). As these values are outside the ranges tested by statisticians in sensitivity analyses (e.g. Marfo & Okyere 2019; Aoki 2020), it is not known whether they present a problem for our analysis. Investigating the sensitivity of meta-analytic models to deviations of this magnitude could be an informative avenue for future research in meta-analysis methodology.

Due to multiple rounds of dataset validation (e.g. Supporting Information 2.10), we do believe that the coalescent data in our analysis represent genuine biological measurements. In our own sensitivity testing, we identified separately the data generated by each statistical method (Figure 2b, Supporting Table 1.6a, Supporting Table 1.6b) and saw no indication that our model parameter estimates, nor their variances, were substantially biased for data from coalescent studies. All models reported in this study met standard model validation criteria (corresponding methods reported under “Methods: Meta-analysis” lines 787-797). In the absence of evidence for statistical bias (which we acknowledge is not evidence of “no” bias), we have retained these data in our overall analysis, to avoid bias that may result from the systematic removal of real data.

Finally, we conducted a systematic investigation into this issue, to ensure that we have not overlooked any standard meta-analytic validation procedures, by reviewing all analyses that cite Marfo & Okyere (2019). As of 12 August 2024, Scopus reports a total of 30 citations of Marfo &

Okyere (2019), of which 26 full texts could be readily obtained and of those 24 were quantitative analyses (published 2020-2024, primarily in health sciences and related fields). All of these cited the aforementioned simulation study in the context of justifying the choice or interpretation of the effect size estimator used, as we have done in our analysis. If the reviewer is aware of a validation protocol that we have overlooked, we would be pleased to receive an example that provides the corresponding methodology, and examine whether it is suitable for our dataset.

We have provided additional information in our manuscript as follows:

- We have summarised the quantitative arguments above, justifying our choice of Hedge's g and our other analytical choices, at Supporting Information 1.4.
- To improve transparency and reproducibility of our dataset, we have added the standard deviations of early and recent estimates for two time-point and coalescent data to the dataset accompanying this paper. These values were previously calculable from the raw data and formulae provided, but their addition to the dataset now makes their subsequent use by readers easier.

Aoki, S. (2020). Effect sizes of the differences between means without assuming variance equality and between a mean and a constant. *Heliyon* 6: e03306

Grissom, R. J. (2000) Heterogeneity of variance in clinical data. *Journal of Consulting and Clinical Psychology* 68: 155-165.

Marfo, P. & Okyere, G. A. (2019). The accuracy of effect-size estimates under normals and contaminated normals in meta-analysis. *Heliyon*. 5: e01838.

Taylor, L. R. (1961) Aggregation, variance and the mean. *Nature* 189: 732-735

b) I still find it difficult to interpret the results of genetic loss as "global"...they may be global for mammals and birds but not for the other groups. I suggest the authors use the word "loss" only when referring to the groups that have actually lost genetic diversity. However, I agree with the authors on the likely scenario that, in the coming decades, the loss of genetic diversity in other groups globally will become more apparent. This suggests that we may be on the brink of a global genetic erosion phenomenon, especially given that significant genetic change is challenging to detect when the median study duration is only six years. I know this comment is a matter of details, but sometimes the devil is in the details.

AUTHOR RESPONSE: In response to this comment, comment by reviewer #4, and the editor's suggestion, we have changed our title to "*Global meta-analysis shows that conservation action is needed to halt genetic diversity loss*". This puts the emphasis on the global nature of the study, rather than implying uniform loss across the globe. Throughout the manuscript, we refer to 'global loss' only when reporting results from the total dataset, and provide further detail about data subsets throughout - including when loss was/was not detected. We appreciate the reviewer's recognition of the nuance in the detection and interpretation of loss, and that more loss is likely in coming decades. We agree that this is a key message of our paper.

c) In an answer to reviewer 4, the authors wrote, "We do not argue that neutral (e.g., microsatellite) genetic diversity loss is a cause of decline. We argue that decline is a cause of genetic diversity loss, and that genetic diversity loss - even neutral diversity - is therefore cause for alarm." I still find this answer quite difficult to follow in terms of its internal logic. The original statement's logic suggests that population decline causes genetic diversity loss and emphasizes concern about this loss. However, it could be perceived as contradictory. In my view, there is a subtle inconsistency. The authors first separate the impact of neutral genetic diversity loss from being a cause of population decline, but then suggest it is alarming. This could be interpreted as contradictory. Or am I misinterpreting the authors here?

AUTHOR RESPONSE: We respectfully suggest that the reviewer has misinterpreted our intended meaning. Anthropogenic activities are driving population decline and biodiversity loss globally (Jaureguiberry et al. 2022). Population decline leads to the loss of both neutral and adaptive genetic diversity, primarily through mechanisms like genetic drift and inbreeding. Thus, a reduction in neutral genetic diversity serves as a key indicator that the population has undergone demographic changes (potentially due to anthropogenic activities and other threats) that are likely to have also led to losses in functional diversity—a concept well justified in the literature (e.g., Kardos et al., 2021).

This means that the cause for alarm is twofold: first, it suggests there that substantial population size declines have occurred across various taxonomic groups and geographic regions (which we show is likely driven by land use change, disease, abiotic natural phenomena, and harvesting/harassment); second, the detected loss of neutral genetic diversity in our study suggests an overall loss of within-species diversity is likely. This loss of genetic diversity could impair species' ability to adapt, maintain fitness, and avoid inbreeding depression, ultimately reducing a population's resilience to abrupt environmental changes, climate shifts, and other challenges, including diseases. We recognise that there has been some confusion around this important message, so now state this more clearly in our introduction as follows (L133-138):

“Quantifying and predicting genetic diversity change over time is essential to biodiversity policy prioritisation, risk assessment, and landscape management^{4,12}. Population decline and fragmentation due to anthropogenic factors, like habitat degradation, unsustainable harvest, invasive species and extreme climatic events¹³⁻¹⁶, lead to genetic erosion¹⁷: loss of genome-wide genetic diversity and adaptive potential. Observed genetic diversity loss is therefore both a signal of population decline, and a conservation concern in its own right⁴.”

Jaureguiberry, P., Titeux, N., Wiemers, M., Bowler, D. E., Coscieme, L., Golden, A. S., Guerra, C. A., Jacob, U., Takahashi, Y., Settele, J., Díaz, S. (2022). The direct drivers of recent global anthropogenic biodiversity loss. *Science advances* 8: eabm9982.

Kardos, M., Armstrong, E. E., Fitzpatrick, S. W., Hauser, S., Hedrick, P. W., Miller, J. M., Tallmon, D. A., Funk, W. C. (2021). The crucial role of genome-wide genetic variation in conservation. *Proceedings of the National Academy of Sciences* 118: e2104642118.

Other minor comments can be found here:

1) Why does the first reference, line 127, start with a 7?

AUTHOR RESPONSE: Nature requires a full-referenced Summary, which appears before the Introduction and contains references 1 – 6.

2) Meta-analysis is first mentioned in line 168. I know the authors may be facing limitations on the number of references they can include, but somehow a reference to a paper on the philosophy and methods for meta-analysis may be useful for readers.

AUTHOR RESPONSE: We have added reference #29 to the introduction to direct readers to more in-depth information about meta-analysis.

Gurevitch, J., Koricheva, J., Nakagawa, S. & Stewart, G., Meta-analysis and the science of research synthesis. *Nature* 555, 175-182, DOI: 10.1038/nature25753. (2018).

3) In line 177, it should read “Using meta-regressions” instead of “Using meta-regression”.

AUTHOR RESPONSE: Changed as suggested

4) I am not sure of the specific guidelines in Nature about including p-values when reporting r, but in line 239, r values are not accompanied by the p-values.

AUTHOR RESPONSE: The corresponding p-values are provided within Supporting Information 1.6 (Supporting Table 1.6b), which is cited next to the r values the reviewer mentions. We do not provide p-values in the main manuscript, as the Results narrative referred to describes a general pattern in r-values, not individual statistics.

5) The header “Disturbance is more common than conservation management” does not fully summarize the main topic or result of the section. More common in what? I assume that is in reporting, but it is not self-explanatory. Please rephrase for clarity.

AUTHOR RESPONSE: The reviewer is correct in their assumption that this title refers to reporting - i.e., disturbances are more commonly reported than conservation management actions are. This is made clear in the subsequent text, e.g., “*Within the temporal timeframe of studies, at least one type of ecological disturbance or conservation management action was reported for 65.11% and 45.75% of the unique populations in our systematic review dataset, respectively*” (L273-275). Given the editorial requirement that section subheadings must be ≤40 characters (including spaces), we prefer to use the subheading “Disturbance more common than management”.

6) In lines 364-365, authors report that “losses of genetic diversity...indicates that the population size declines at the basis of these genetic diversity losses must be considerable”. The use of “must” here feels a bit strong. I agree with reviewers that in many cases, population size and genetic diversity correlate, but this is not always true. I think the verb “may” would be more appropriate.

AUTHOR RESPONSE: We have changed “must be” to “are likely to be”.

7) In line 527, for improving clarity, can the authors write that studies were suitable for inclusion in our analysis only if they satisfied ALL the following criteria?

AUTHOR RESPONSE: Changed as suggested

8) In figure 1, I find it difficult to relate “How we measure change” and “Where and What we measure” with the text in the boxes below those headers...I would expect to find answers to those questions (i.e., what taxonomic groups, realms, and metrics are part of the study)

AUTHOR RESPONSE: We have changed these subtitles from questions to statements to better describe the text in each box. For example, under “How we measure change”, the subtitle: “Does study design influence genetic diversity change?” now reads: “*Study design variables that may influence genetic diversity change*”. Under “Where and what we measure”, the subtitle: “Does population context influence genetic diversity change?” now reads: “*Population context variables that may influence genetic diversity change*”. We have also updated the threat and conservation statements accordingly for consistency.

9) In Figure 2g, Abiotic Phenomenon and Human Natural are not aligned with a specific row...this is confusing. Please clarify and redraw if needed.

AUTHOR RESPONSE: We have revised all figures to adhere with Nature’s editorial guidelines, and have redrawn these labels in the newly revised Figure 3.

10) In Figure 1g and in Extended Figure 6, there are blue colors that are too similar to each other, making it difficult to understand the figures. Please improve color legends for easier understanding of the figures.

AUTHOR RESPONSE: Figure 3 and Extended Figure 7 both use a palette of 12 colours that were chosen to optimise clarity and contrast for colour-blind readers. We note that on both figures, the data bars and key/labels are presented in a consistent order to further facilitate interpretation. If our manuscript is accepted, we welcome advice from the Nature publishing team about the colours used in our figures.

Referee #2 (Remarks to the Author)

I am satisfied with the authors' revision and have no further comments.

AUTHOR RESPONSE: We thank the reviewer for re-examining our manuscript and their positive assessment.

Referee #3 (Remarks to the Author)

I have had the opportunity to review an earlier version of the manuscript titled "Conservation action is needed to halt pervasive genetic diversity loss". In this ms the authors perform a meta-analysis where they evaluate the effect of different human activities and natural phenomena on genetic diversity over time. They also test the effect of different conservation actions on genetic diversity. The authors also incorporate phylogenetic data to account for the potential effect of shared evolutionary history on genetic diversity.

The results of the present analyses suggest that that most of the populations are experiencing loss of genetic diversity and that that pattern is widespread and significant across taxa and ecosystems. The authors also show that less than half of the studied species were subject to conservation actions despite pervasive genetic diversity loss. Genetic diversity loss is mainly driven by land-use change, disease, abiotic natural phenomena, harvesting, and harassment. The authors find that only supplementation had a significant and positive impact on genetic diversity.

In this reviewed version of the manuscript the authors have made a great effort to address my comments and those of additional 3 reviewers. In my opinion they have carried out a thorough revision and they have address satisfactory all points that I have raised in my comments. At that stage I do not have any further comments and I am happy to recommend the manuscript for publication.

AUTHOR RESPONSE: We thank the reviewer for re-examining our manuscript and their positive assessment.

Referee #4 (Remarks to the Author)

The authors have provided a thorough and thoughtful response to all of the critiques and points of confusion presented by the reviewers. I'm not qualified to comment on all of the technical points of analysis, but the authors appear to have considered each criticism carefully, with responses that strike a balance between working with the limitations of the data and extracting from them as much as possible. This large, comprehensive, and thorough study will be of interest to many people in the field.

AUTHOR RESPONSE: We thank the reviewer for re-examining our manuscript and their positive assessment.

I still struggle to fully appreciate the real-world implications of the results, and swapping in “global” for “pervasive” seems mostly cosmetic, but I can see how there is room for different views here. The authors clarify that they can’t point causal arrows from disturbances or conservation action to genetic diversity change (multiple, very different interpretation are plausible; point 2 in initial review), and that they basically see genetic diversity decline as an indication of population decline rather than the reverse (point 7). The latter admission would seem clash with statements at the very core of the study, such as “Mitigating genetic diversity loss is a major global challenge” or that we need to “halt genetic diversity loss”, given that genetic diversity is mostly a proxy for the real quantity of interest, population size, on which there is lots of current study and debate.

AUTHOR RESPONSE: We appreciate that the reviewer is open to different views on this topic. In light of similar concerns raised by Reviewer #1, we have revised the title to emphasise the global nature of the study, rather than implying uniform global loss (see response to comment ‘b’ by Reviewer #1).

We agree that population decline is the key driver of genetic diversity loss. However, our study does not present genetic diversity merely as a proxy signaling ecological processes. Instead, we emphasise the intrinsic importance of population-level genetic diversity, which underpins adaptive potential, population resilience, and overall population health. In this sense, genetic diversity is the primary quantity of interest, as it directly relates to the capacity for populations to survive and adapt in changing environments.

The reviewer’s perspective aligns with a common approach in conservation science, which understandably emphasises immediate ecological threats by prioritising short-term actions to avert species extinction (Frankham 2010), often at the expense of considering long-term evolutionary processes. While ecological interventions are crucial, we argue that integrating evolutionary concepts into conservation management (i.e., long-term thinking grounded in genetic diversity and adaptive potential) is essential for achieving enduring conservation outcomes (Cook & Sgrò 2017).

Our study advocates for managing genetic diversity loss not merely as a consequence of population decline, but as a critical conservation target in its own right. This necessitates ecologically and evolutionarily informed management strategies that address the causes and consequences of genetic diversity loss, including disturbances leading to population decline, fragmentation, and isolation. We recognise that there has been some confusion around this important message, so now state this more clearly in our introduction as follows (L133-138):

“Quantifying and predicting genetic diversity change over time is essential to biodiversity policy prioritisation, risk assessment, and landscape management^{4,12}. Population decline and fragmentation due to anthropogenic factors, like habitat degradation, unsustainable harvest, invasive species and extreme climatic events¹³⁻¹⁶, lead to genetic erosion¹⁷: loss of genome-wide genetic diversity and adaptive potential. Observed genetic diversity loss is therefore both a signal of population decline, and a conservation concern in its own right⁴.”

Cook, C.N. & Sgrò, C.M. (2017), Aligning science and policy to achieve evolutionarily enlightened conservation. *Conservation Biology* 31: 501-512.

Frankham R. (2010). Challenges and opportunities of genetic approaches to biological conservation. *Biological Conservation* 143: 1919–1927.

RESPONSE, final revision

Referee #1 (Remarks to the Author):

Dear Dr. Catherine E. Grueber and co-authors,

Many congrats for this fantastic effort and piece of research. I also appreciate your willingness to enhance the paper based on the reviewer's recommendations. I don't have any further comments or suggestions, aside from a few minor typos I noticed (see below). I'm looking forward to seeing this paper published.

RESPONSE: we thank the reviewer for their support in improving this work.

Line 170 "published records"..., maybe better "published papers"? Or "studies"?

RESPONSE: here we are referring to the number of records identified in the search databases, we have made a minor edit to the sentence to clarify this.

Line 175 - "across a range of geographic regions, timeframes"...should be time frames

RESPONSE: we have made this correction throughout the main and supporting text.

"Our study provides the most comprehensive and impartial investigation of within-population genetic diversity change to date"...what do you mean by impartial? Free of biases??? I would avoid the use of adjectives to self-qualify the robustness of your study...

RESPONSE: to avoid ambiguity, we have removed the word "impartial".

In extended figure 7...change "% where disturbance reported" by "% where disturbance is reported"

RESPONSE: changed as suggested in the insets of both panels b) and d).

Yours sincerely,
Dr. David Nogués-Bravo